PLOS One logo

## RESEARCH ARTICLE

# Enhancing rainfall estimation accuracy with machine learning, cloud masking, and multi-source data: A case study of four coastal provinces in central Vietnam

Dong Vu Duy[1], An Hung Nguyen 🔾[1]*, Phat T. Nguyen[1], Huyen Thi Nguyen[1], Thanh Thi Nhat Nguyen[2]

**1** Faculty of Radio-Electronic Engineering, Le Quy Don Technical University, Hanoi, Vietnam, **2** VNU University of Engineering and Technology, Vietnam National University, Hanoi, Vietnam

* hungan@lqdtu.edu.vn

## Abstract

This study develops the LGBM-3SC-CF product, a machine learning framework that integrates multi-source data (Himawari-8, ERA5, ASTERDEM, and rain gauge) to significantly improve rainfall classification and estimation accuracy for four coastal provinces in Central Vietnam. Methodologically, our core contribution lies in the proposed 3-stage classification architecture combined with a novel cloud masking technique. To address the severe class imbalance inherent in rainfall datasets, we further employed an effective data balancing technique based on rainfall intensity distribution alongside feature selection based on cloud- and rain-forming factors. The performance of the proposed rainfall product was compared with four existing regional rainfall products: IMERG Final Run V07, IMERG Early Run V06, GSMaP_MVK_Gauge V07, and PERSIANN-CCS. The LGBM-3SC-CF achieved the highest performance, with a Critical Success Index (CSI) of 0.55 and a Probability of Detection (POD) of 0.74, and achieved the Correlation Coefficient (CC) of 0.47 and the Modified Kling-Gupta Efficiency (mKGE) of 0.47. Furthermore, it obtained the lowest error metrics, with a Mean Absolute Error (MAE) of 2.66 mm/h and a Root Mean Squared Error (RMSE) of 5.48 mm/h. This study not only establishes a robust machine learning framework but also provides the essential methodological foundation for developing near real-time rainfall estimation models through the seamless substitution of the atmospheric reanalysis features with near real-time meteorological features.

## Introduction

The Central region of Vietnam is frequently affected by extreme weather events closely associated with rainfall variability, including floods and landslides [1]. Consequently, the development of a high-precision rainfall dataset plays a crucial role

**Data availability statement:** All dataset files are available from the Figshare database at https://doi.org/10.6084/m9.figshare.30782003.

**Funding:** The author(s) received no specific funding for this work.

**Competing interests:** The authors have declared that no competing interests exist.

in supporting economic development, improving the accuracy of forecasting models, and enabling the formulation of more effective disaster prevention and mitigation strategies [2].

Various approaches have been employed for rainfall estimation, including rain gauge stations, weather radar, and meteorological satellites [3,4]. Nevertheless, each of these methods has inherent limitations in terms of spatial coverage, temporal resolution, terrain influence, and sensor quality. In recent years, the integration of machine learning (ML) and deep learning (DL) techniques with multi-source data has emerged as a promising solution to improve the accuracy of rainfall estimation. ML models, in particular, are capable of effectively handling moderately sized datasets and capturing complex, nonlinear relationships between input features and rainfall output, while requiring fewer computational resources and being less complex than DL models. As a result, ML methods are increasingly favored in rainfall estimation research [5,6].

Enhancing the accuracy of rainfall estimation fundamentally necessitates a robust preliminary rainfall classification [7]. Rainfall classification approaches in the literature are generally divided into two main categories: (i) single-stage classification and (ii) two-stage classification. In the single-stage approach, a single model is used to classify the input data into rain/no-rain or directly into various rainfall intensity classes, followed by rainfall estimation [8]. In the two-stage approach, the first model classifies the data into rain and no-rain regions, while the second model further classifies the rain regions into different rainfall intensity classes, and rainfall estimation is then performed based on these classes [5]. Nevertheless, regardless of the chosen approach (single-stage or two-stage), rainfall distribution imbalance persistently remains a formidable challenge for accurate modeling. Therefore, this study proposed a multi-stage approach, which divides the rainfall range into several sequential sub-intervals and performs binary (two-class) classification for each interval to effectively mitigate sample imbalance and significantly improve the model's predictive accuracy on minority classes.

In addition to architectural designs, a variety of data balancing techniques — primarily relying on numerical and mathematical processing— have been proposed to mitigate the effects of class imbalance in classification tasks [9]. These often involve strategies like class weighting (CW), random oversampling of minority classes, and synthetic oversampling methods such as the Synthetic Minority Over-sampling Technique (SMOTE) [10], Borderline SMOTE [11,12], and Adaptive Synthetic Sampling (ADASYN) [12]. For instance, in [13], three ML models, including RF, K-Nearest Neighbors (KNN), and Decision Tree (DT), were used to estimate rainfall in India, with SMOTE being applied to handle the associated data imbalance. Among the models, RF combined with data balancing yielded the best classification results, with an F1-score of 0.91, Precision of 0.89, Recall of 0.94, and Accuracy of 0.91. Similarly, in [14], the XGBoost (XGB) model was applied in combination with SMOTE and SMOTEN techniques to balance daily rainfall classification data. The results showed that the XGB model using SMOTEN achieved the highest accuracy of 92.92%, followed by SMOTE at 90.58%, while the unbalanced model achieved only 75.36%. However, each balancing method has its own strengths and limitations; thus, its

selection should depend on the dataset's characteristics. This is particularly important for multi-class tasks with imbalanced classes, such as heavy or very heavy rainfall with very few samples [15,16]. Therefore, selecting and combining different data balancing techniques that are suitable for the distribution characteristics of each rainfall class can potentially enhance the accuracy for minority classes in particular, as well as improve the overall performance of rainfall classification models in general [17]. This is also an approach utilized in this paper.

Furthermore, a major challenge in rainfall classification stems from the severe data imbalance, often resulting in the misclassification of non-rain samples as rain events (false alarms), thereby requiring effective spatial filtering to suppress this rain over-detection phenomenon. Addressing this, cloud masks—derived from key atmospheric properties like Cloud Top Height (CTH), Cloud Top Temperature (CTT), and Cloud Phase (CP)—are incorporated as a crucial strategy to constrain the rainfall estimation area, significantly improving the reliability and accuracy of the detection system [18–20]. A common methodology for developing these masks relies on analyzing Brightness Temperature (BT) thresholds and Brightness Temperature Differences (BTDs) derived from infrared (IR) channels of satellite imagery, such as Himawari-8. For example, the study in [22] employed various thresholds of BTs and BTDs from Himawari-8 IR channels to classify clouds into types based on their relationship with surface rainfall. This approach, validated against GPM satellite products, achieved high overall accuracy (0.97 for cloud/no-cloud classification and 0.72 for rain-cloud classification). Similarly, the research in [23] focusing on specific regions, such as southern China during 2020–2021, applied the BT threshold of 255 K from the B14B channel (11.2 $\mu$m) of Himawari-8 to successfully detect convective rain clouds. However, the critical challenge remains that the appropriate BT threshold for accurately identifying rain-bearing clouds is highly dependent on local climate, meteorological, and topographical conditions. This study customizes the cloud masking process by analyzing the statistical distribution characteristics of the dataset specific to the research area, thereby selecting appropriate thresholds for generating cloud masks applied within the proposed rainfall classification model. Also, accurate rainfall classification and estimation derived from satellite data fundamentally requires the integration of local features, such as topography and regional meteorological patterns, as rainfall on the Earth's surface is closely related to these conditions [21,22]. Consequently, this study utilizes ERA5 and ASTER DEM data specific to the research area to enhance the accuracy of rainfall estimation and classification. To optimize model performance and avoid redundancy, careful feature selection from these multi-source features is employed.

In summary, this study aims to improve the accuracy of rainfall classification and estimation—addressing the aforementioned methodological challenges—for four coastal provinces in the Central region of Vietnam during 2019–2023. This is achieved by utilizing the LGBM model integrated with multi-source data, including Himawari-8 satellite imagery, ERA5 reanalysis data, ASTER DEM, and regional rain gauge networks. The main contributions of this study are as follows:

i. A novel three-stage rainfall classification architecture is proposed. In Stage 1, the input data are classified into rain and no-rain regions. In Stage 2, the rain regions identified in Stage 1 are further classified into low-intensity rain and high-intensity rain. In Stage 3, the low-intensity rain areas are classified into small rain and moderate rain, while the high-intensity rain areas are classified into heavy rain and very heavy rain. This hierarchical structure enhances the overall classification performance of the proposed rainfall product, especially for the heavy rain and very heavy rain classes.

ii. A feature selection strategy is proposed to identify relevant features that effectively reflect the topographical and meteorological characteristics of the study area, with the goal of improving the performance of classification and regression models.

iii. The proposed use of cloud masks aims to enhance the accuracy in distinguishing between rain-bearing and no-rain-bearing cloud regions, as well as between clouds associated with low-intensity and high-intensity rainfall. Thereby improving the overall rainfall classification performance of the proposed precipitation product.

iv. A strategy is proposed to combine different data balancing techniques to improve rainfall classification performance. Specifically, range-based rainfall (RR) augmentation is applied to the rain/no-rain classification model, while the class weight (CW) technique is used for the classification of low- and high-intensity rain, as well as for distinguishing between small and moderate and heavy and very heavy rainfall.

The remainder of this paper is organized as follows: Sect 2 presents the data and methodology. Sect 3 discusses the experimental results and evaluation. Sect 4 provides conclusions and potential directions for future research.

## Data and methods

### Case study

The case study is the four coastal provinces located in the Central region of Vietnam, including Quang Binh, Quang Tri, Thua Thien Hue, and Da Nang (Fig 1). The terrain in this region is predominantly mountainous, with an average elevation of approximately 500 meters. It borders the East Sea to the east and Laos to the west, separated by the steep and high Truong Son mountain range. Moist air masses from the sea move inland, where they are forced to rise upon encountering elevated terrain, resulting in orographic rainfall on the windward slopes. Conversely, hot and dry air masses blowing from the inland areas toward the coast have low humidity, creating the phenomenon known as the *Lao wind*. The region's narrow topography, combined with complex and continuously changing terrain, leads to intricate and unpredictable patterns of air movement. The climate is characterized by a monsoonal regime, influenced by the northeast monsoon and the southwest foehn wind [23].

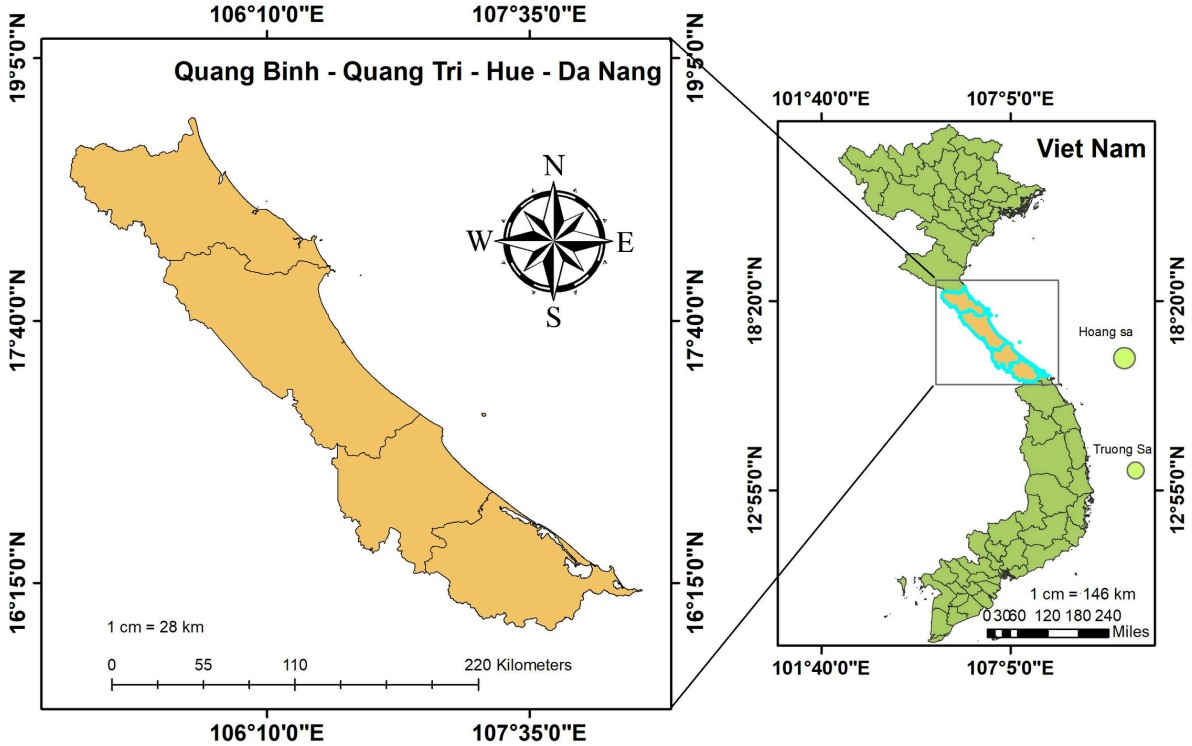

**Fig 1. Study area.**

## Datasets

The input dataset includes rain gauge data, Himawari-8 satellite data (BT), and auxiliary data comprising ERA5 reanalysis and ASTER DEM,all collected during 2019–2023. The Himawari-8 data, provided by the Japan Meteorological Agency (JMA), have a temporal resolution of 1 hour and a spatial resolution of 2 km [5]. In this study, Brightness Temperature (BT) data from 10 individual Infrared (IR) channels—namely I4B, WVB, B09, B10, B11, B12, IRB, B14, I2B, and B16—were utilized, alongside 45 derived dual-channel IR features (calculated as the BT differences between these individual IR channels). This comprehensive collection results in a total of 55 Himawari-8 features.

The ERA5 dataset, developed by the European Centre for Medium-Range Weather Forecasts (ECMWF), has a temporal resolution of 1 hour and a spatial resolution of 25 km [24]. In this study, 17 features from ERA5 were utilized, including the K-index (KX), Convective Inhibition (CIN), Convective Available Potential Energy (CAPE), Total Column Water (TCW), Total Column Water Vapour (TCWV), Instantaneous Moisture Flux (IMF), Anisotropy of Sub-gridscale Orography (ISOR), Slope of Sub-gridscale Orography (SLOR), Relative Humidity (R) at 250 hPa, 500 hPa, and 850 hPa, U-component of wind (UWIND) at 250 hPa, 500 hPa, and 850 hPa, and V-component of wind (VWIND) at 250 hPa, 500 hPa, and 850 hPa. Among these, the proposed rainfall product employs a common feature set, comprising TCW, SLOR, R850, UWIND850, and VWIND850, for training all models from M1 to M8 within the proposed architecture. Meanwhile, the rainfall product utilizing the RF importance technique applies all 17 features to identify distinct subsets of important features for each individual model. Besides the ASTER DEM data, developed by NASA, it provides global-scale elevation data at a spatial resolution of 30 meters [25].

The comparative data consist of five regional rainfall products, including IMERG products (Early Run and Final Run) developed by NASA and JAXA, which provide precipitation estimates at a spatial resolution of 0.1° × 0.1° and a temporal resolution of 30 minutes. IMERG Early Run (V06) is a near-real-time product with a latency of approximately 6 hours [26], while IMERG Final Run (V07) is a delayed product with a latency of about 3.5 months and has been bias-corrected using surface rain gauge data. GSMaP_MVK_Gauge (V07), developed by JAXA, offers hourly precipitation data at a spatial resolution of 0.1° × 0.1° with a latency of approximately 3 days [27]. PERSIANN_CCS, developed by the Center for Hydrometeorology and Remote Sensing (CHRS), is a near-real-time product with high spatial resolution (0.04° × 0.04°), hourly temporal resolution, and a latency of about 30 minutes [28]. In this study, both the input datasets and the comparative datasets were resampled to a common spatial resolution of 4 km and a temporal resolution of 1 hour.

Lastly, rain-gauge data collected from 175 automatic weather stations (AWS) across the region were used both as target labels for model training and as ground truth for evaluating the performance of the rainfall products. The total rainfall accumulated in the study area during the 2019–2023 period is depicted in Fig 2. In addition, weather radar images collected during the 2019–2023 period by the National Center for Hydro - Meteorological Forecasting (NCHMF) were

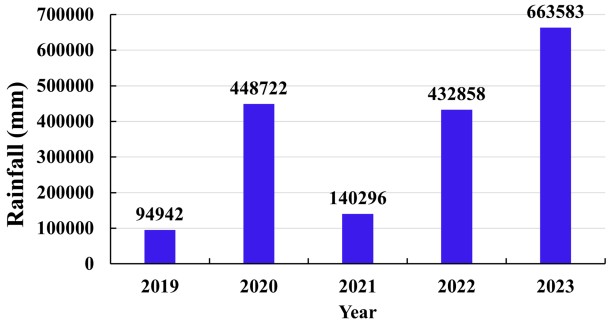

**Fig 2**. Annual accumulated rainfall in the study area.

used to visually assess the spatial consistency between the rainfall maps of the proposed product and the radar-observed rainfall maps during several rainfall events.

## Methodology

**Framework.** This study proposes a machine learning framework for detailed rainfall classification and regression, as illustrated in Fig 3. Three machine learning algorithms were investigated in this study: LGBM, XGB, and RF. The input

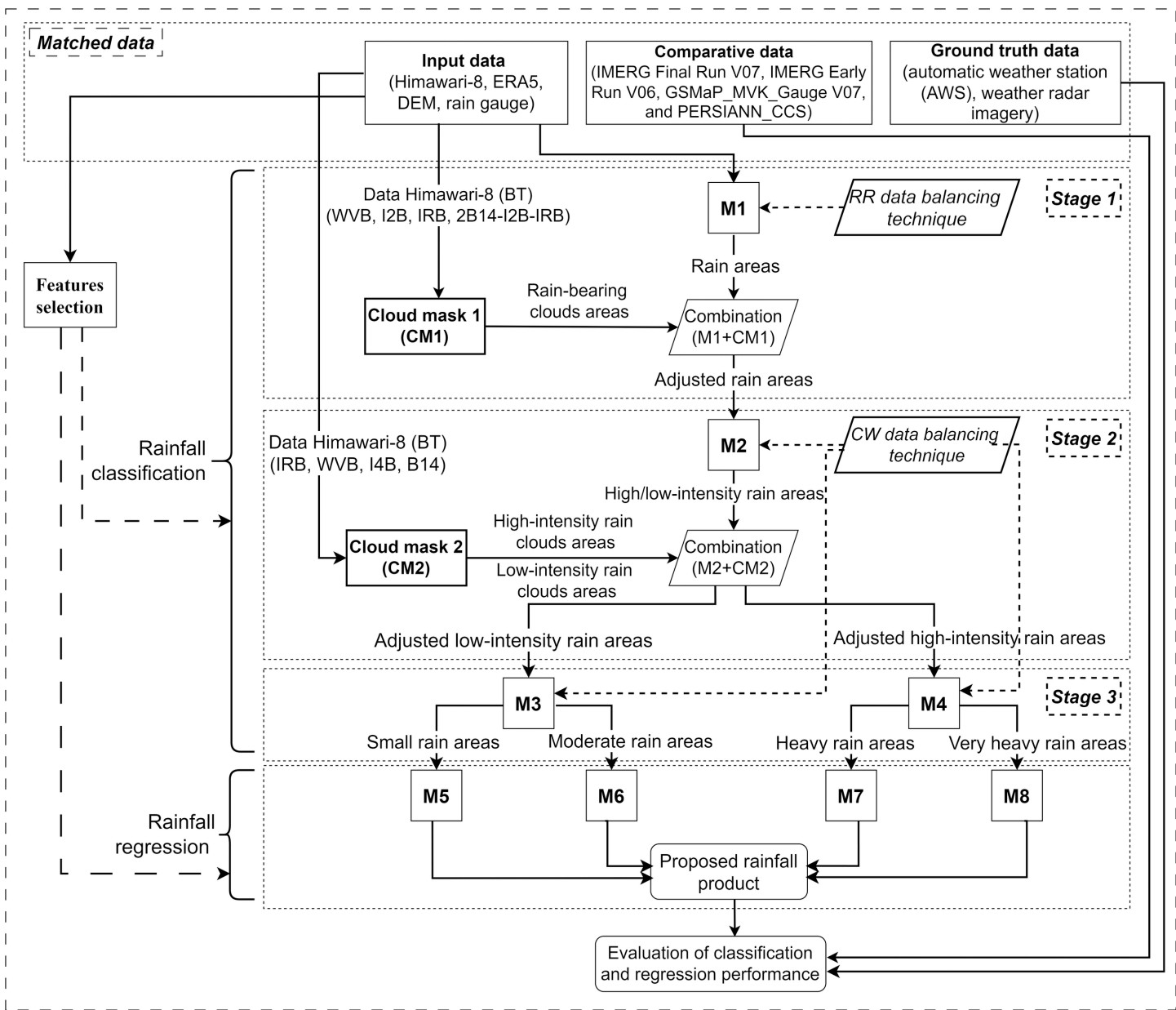

**Fig 3. Proposed framework for rainfall estimation.**

data are collected from multiple sources with varying spatial and temporal resolutions, as described in Sect 2.2. Therefore, before training, these datasets are standardized to achieve a consistent spatial resolution of 4 km and a temporal resolution of 1 hour. The overall methodology of this study is structured into three main components: a three-stage rainfall classification process, a subsequent rainfall regression (estimation) process, and a comprehensive performance evaluation procedure. The detailed steps for each process are described in the following sections.

First, the three-stage classification process is described below:

Stage 1: Performs the initial rain/no-rain detection through a logical combination of two results. Model M1 provides the preliminary classification of rain/no-rain areas. Simultaneously, Cloud Mask 1 (CM1), generated from Himawari-8 BT thresholds, identifies rain-bearing and non-rain-bearing clouds. The final rain area is defined only by regions that are simultaneously classified as rain by M1 and rain-bearing by CM1. Regions classified as rain by M1 but falling outside the rain-bearing mask are reclassified as no-rain.

Stage 2: Performs the low/high-intensity rainfall classification using the combined rain areas identified in Stage 1 as input. Model M2 provides the preliminary low/high intensity classification. Concurrently, Cloud Mask 2 (CM2)—derived from specific BT thresholds—distinguishes between clouds producing low-intensity and high-intensity rainfall. The final high-intensity rainfall area is defined only by regions that are simultaneously classified as high-intensity by M2 and identified as high-intensity rain-bearing by CM2, while the remaining areas are then considered low-intensity rainfall.

Stage 3: Refines the rainfall intensity classification through a parallel branching process based on the output of Stage 2 (M2 + CM2). Areas previously classified as low-intensity rainfall are fed into Model M3 for detailed small/moderate rain classification. Concurrently, areas classified as high-intensity rainfall are input into Model M4 for detailed heavy/very heavy rain classification. The specific rainfall thresholds used to define these fine-grained intensity
levels for the input data of both M3 and M4 are detailed in Table 1.

Second, after the rainfall was classified into four categories, including small, moderate, heavy, and very heavy, the corresponding samples were transferred to models M5 through M8 to perform rainfall regression. Finally, the final rainfall product is generated by integrating the outputs of these four models, forming the rainfall map of the research area. The classification and regression performance of the proposed rainfall product was compared with four existing regional rainfall products, namely GSMaP_MVK_Gauge V07, IMERG Final Run V07, IMERG Early Run V06, and PERSIANN_CCS—through quantitative evaluation metrics against rain gauge observations and qualitative visual assessment against radar imagery.

Additionally, to optimize model complexity and efficiency, feature selection was implemented for both the classification and regression models to reduce the number of input variables. This study deployed two feature selection approaches: one utilizing a fixed feature set applied across all classification and regression models, and the second where each model employs the Random Forest (RF) Importance technique for feature selection. Details of these two feature selection schemes are presented in Sect 3.1.1. Also, to enhance classification quality and address class imbalance inherent in rainfall data, two data balancing techniques were employed. Specifically, ange-based Rainfall (RR) technique was applied to

**Table 1**. Rainfall thresholds used to classify rainfall classes.

| Rainfall class | Rainfall threshold (mm/h) |
|---|---|
| Small | 0.1 − 1.0 |
| Moderate | 1.0 − 5.0 |
| Heavy | 5.0 − 30.0 |
| Very heavy | >30.0 |

Model M1 by randomly increasing the number of minority class samples (Sect 3.1.2, while Class Weight (CW) was utilized for Models M2, M3, and M4.

**Mask cloud generation.** Two dedicated cloud masks were integrated into the proposed rainfall product to significantly enhance classification performance at different stages of the process. The first, Cloud Mask 1 (CM1), was utilized in conjunction with the output of the initial rain/no-rain classification model (M1) to filter false detections. Subsequently, Cloud Mask 2 (CM2) was applied alongside the results of the Low-Intensity/High-Intensity classification model (M2) to refine the distinction between rainfall intensity levels.

CM1 relies on BT values from Himawari-8 channels (WVB, IRB, I2B, B14) and derived BT differences (B14 − I2B, B14 − WVB, and $2 \times$ B14 − I2B − IRB) as input features. These features were selected because they are physically sensitive to cloud properties highly correlated with rain-bearing clouds [29–31]. The procedure for determining the optimal thresholds for CM1 is an iterative process executed in three main steps:

Step 1: Establishes the initial BT range for each individual input channel and difference based on the actual rain points derived from the rain gauge stations.

Step 2: Performs binary classification. A point is classified as a rain-bearing cloud only if its BT value in every single input channel falls within the defined range established in Step 1; otherwise, it is classified as a non-rain bearing cloud.

Step 3: Involves performance evaluation and iterative adjustment. A confusion matrix is computed, focusing on maximizing True Positives (TP) and minimizing False Negatives (FN). If the performance objective is not met (e.g., FN is too high and TP is too low), the BT range is iteratively expanded at both ends using discrete empirical values, and Step 2 is repeated until the desired TP/FN balance (e.g., FN $\approx 34\%$, TP $\approx 66\%$) is achieved, ensuring the stable detection of rain-bearing cloud regions.

CM2 is generated analogously to CM1, but with a different set of input features (WVB, IRB, I4B, and B14) tailored to distinguish low-intensity from high-intensity rain clouds.

**Algorithm.** LGBM is an algorithm belonging to the Gradient Boosting Decision Tree (GBDT) family, introduced by Microsoft in 2017 [32]. It operates based on the principle of combining multiple weak learners to form a strong learner, thereby improving overall predictive performance. Its objective is to iteratively minimize the first-order derivative (gradient) of the loss function in each boosting round [32]. Mathematically, the LGBM model is initialized with a constant value that minimizes the overall error across the entire training dataset.

Given a training dataset consisting of $n$ samples $\{(x_i, y_i)\}_{i=1}^n$, where: $x_i \in \mathbb{R}^d$ is the feature vector of the $i$-th sample, and $y_i$ is the true label of the $i$-th sample. The model is constructed as an ensemble of $M$ decision trees (DTs) in the following form:

$$\hat{y}_i = \sum_{k=1}^{M} f_k(x_i), \quad f_k \in \mathcal{P} \tag{1}$$

Where, $\hat{y}_i$ is the predicted value of the $i$-th sample, $f_k$ is the $k$-th decision tree, and $\mathcal{P}$ denotes the set of all regression trees in the model. Mathematically, the LGBM model is initialized with a constant value that minimizes the overall error across the entire training dataset:

$$f_0(x) = \arg\min_{\gamma} \sum_{i=1}^{n} L(y_i, \gamma) \tag{2}$$

For classification problems, the loss function $L(y_i, \gamma)$ takes the form:

$$L(y_i, \gamma) = - [y_i \log(\gamma) + (1 - y_i) \log(1 - \gamma)] \tag{3}$$

For regression problems, the loss function $L(y_i, \gamma)$ is defined as:

$$L(y_i, \gamma) = (y_i - \gamma)^2 \tag{4}$$

The model is trained with $M$ trees, and in each iteration $m \in \{1, \ldots, M\}$, the following basic steps are performed:

**Step 1:** Compute the derivative of the loss function for each sample $i$. This represents the residual ($r_i^{(m)}$) that the model aims to correct in order to reduce the prediction error:

$$r_i^{(m)} = \left[ \frac{\partial L\left(y_i, f_{m-1}(x_i)\right)}{\partial f_{m-1}(x_i)} \right], \quad i = (1, \ldots, m) \tag{5}$$

**Step 2:** Train a new tree to approximate the residual errors, where $h_m(x)$ is the decision tree trained at iteration $m$:

$$h_m(x) = r_i^{(m)} \tag{6}$$

**Step 3:** Apply a learning rate $\mu$ ($\mu \in (0, 1]$) to update the model $f_m(x)$ using the tree trained at iteration $m$:

$$f_m(x) = f_{m-1}(x) + \mu h_m(x) \tag{7}$$

After $M$ iterations, the final prediction for a sample is:

$$f_M(x) = f_0(x) + \sum_{m=1}^{M} \mu h_m(x) \tag{8}$$

LGBM employs histogram-based algorithms and a leaf-wise tree growth strategy, combined with two key techniques: Exclusive Feature Bundling (EFB) and Gradient-based One-Side Sampling (GOSS). EFB reduces the number of features by removing non-correlated features, thereby shrinking the feature space and accelerating training. GOSS retains instances with large gradients, reducing the number of training samples while preserving critical information, which enhances both model performance and accuracy [33].

XGBoost (XGB) is a decision tree-based optimization technique that builds on the gradient descent method. It constructs an ensemble of weak learners derived from the original dataset to minimize a predefined loss function and improve predictive performance [34]. XGB employs a level-wise tree growth strategy (breadth-first expansion), where each subsequent tree is trained to correct the residuals (errors) of the preceding trees, thereby iteratively enhancing the overall model accuracy [35]. The XGB algorithm has been demonstrated to achieve high performance in classification and regression tasks with big datasets [36].

The Random Forest (RF) algorithm, first introduced by Breiman in 2001, constructs an ensemble of decision trees using the bootstrap aggregation (bagging) method [37]. The decision trees are randomly sampled from the input data and trained in parallel to optimize the learning process. The final prediction is obtained through majority voting for classification tasks or averaging for regression tasks [8]. During training, decision trees are grown level by level. Deeper trees provide more detailed learning capacity and potentially higher accuracy; however, they are also more prone to overfitting and require greater computational resources [38].

**Training and evaluation.**

*a) Training*

The dataset used for training the models accounts for 80% of the total original data. In this study, the *k*-fold cross-validation technique ($k = 5$) is employed for model training and validation to prevent overfitting.

This study constructed an evaluation dataset based on a spatiotemporal sampling framework, which captures the continuous variability of rainfall across both spatial and temporal scales. Specifically, observations from each station within the study area, recorded at five specific time points each day (00:00, 06:00, 12:00, 16:00, and 21:00) during the years 2019–2023, were designated for use in the evaluation and ground truth dataset. Rainfall data corresponding to all other time points were used to construct the training dataset. Table 2 provides a detailed summary of the main hyperparameters employed in training the LGBM, XGB, and RF models for both rainfall classification and regression.

The RR data balancing technique was used for model M1 (rain/no-rain classification). Specifically, the original rainfall range from 0.1 to 95.2 mm/h (the recorded maximum rainfall intensity) was divided into 11 smaller intervals: 0.1–1.0 mm/h, 1.0–2.0 mm/h, 2.0–3.5 mm/h, 3.5–5.0 mm/h, 5.0–8.0 mm/h, 8.0–12.0 mm/h, 12.0–20.0 mm/h, 20.0–30.0 mm/h, 30.0–40.0 mm/h, 40.0–50.0 mm/h, and >50.0 mm/h. New synthetic samples are randomly augmented in each rainfall interval at varying ratios, following the principle that higher rainfall intervals receive proportionally more samples than lower ones. This helps balance the distribution within each rain class and mitigates the imbalance between the abundant small rain class (0.1–1.0 mm/h) and the minority rain classes.

Meanwhile, the CW technique was applied to balance the data for the rainfall classification models M2, M3, and M4, in which higher weights were assigned to the minority class. The CW values were calculated based on the degree of class imbalance in the input dataset for each model. The number of rainfall samples for each class in the input datasets of models M1–M4 is illustrated in Fig 4.

*b) Evaluation*

The dataset used for evaluation accounts for 20% of the original dataset and is independent from the training dataset. This study evaluates the classification performance of models M1 to M4 under both balanced and imbalanced data conditions in order to assess the effectiveness of data balancing techniques on the classification results of these models.

To assess the classification and regression performance of the two proposed rainfall products corresponding to two feature selection strategies, the comparisons are conducted against the four regional rainfall products using rain gauge observations as reference. Additionally, radar imagery is used as a reference in selected rainfall events for both qualitative and quantitative assessments of the classification and regression performance of the proposed rainfall products. In addition, the proposed rainfall product with the three-stage rainfall classification architecture was compared with single-stage and two-stage rainfall classification products using the same machine learning model (LGBM).

Several performance metrics for the classification models are summarized in Table 3. These include: Probability of Detection (POD) indicates the model's ability to detect actual rainfall occurrences. The Critical Success Index (CSI) represents the proportion of correctly predicted rainfall samples out of all samples classified as rain. The Equitable Threat Score (ETS) measures the accuracy of predictions for rain events while accounting for random hits. The Heidke Skill Score (HSS) evaluates the overall classification accuracy across both rain and non-rain classes. The BIAS indicates the model's tendency to overestimate or underestimate rainfall occurrences (BIAS > 1 implies overestimation, BIAS < 1 implies underestimation). The False Alarm Ratio (FAR) is the proportion of falsely predicted rain events to the total number of predicted rain events. Therefore, 1 − FAR reflects the precision of rainfall classification.

**Table 2**. List of hyperparameters used for model training.

| STT | Hyperparameters name | |
|-----|----------------------|-----|
|     | **Classification model** | **Regression model** |
| 1 | n_estimators | n_estimators |
| 2 | max_depth | max_depth |
| 3 | subsample | subsample |
| 4 | colsample_bytree | colsample_bytree |
| 5 | learning_rate | learning_rate |
| 6 | class_weight | |

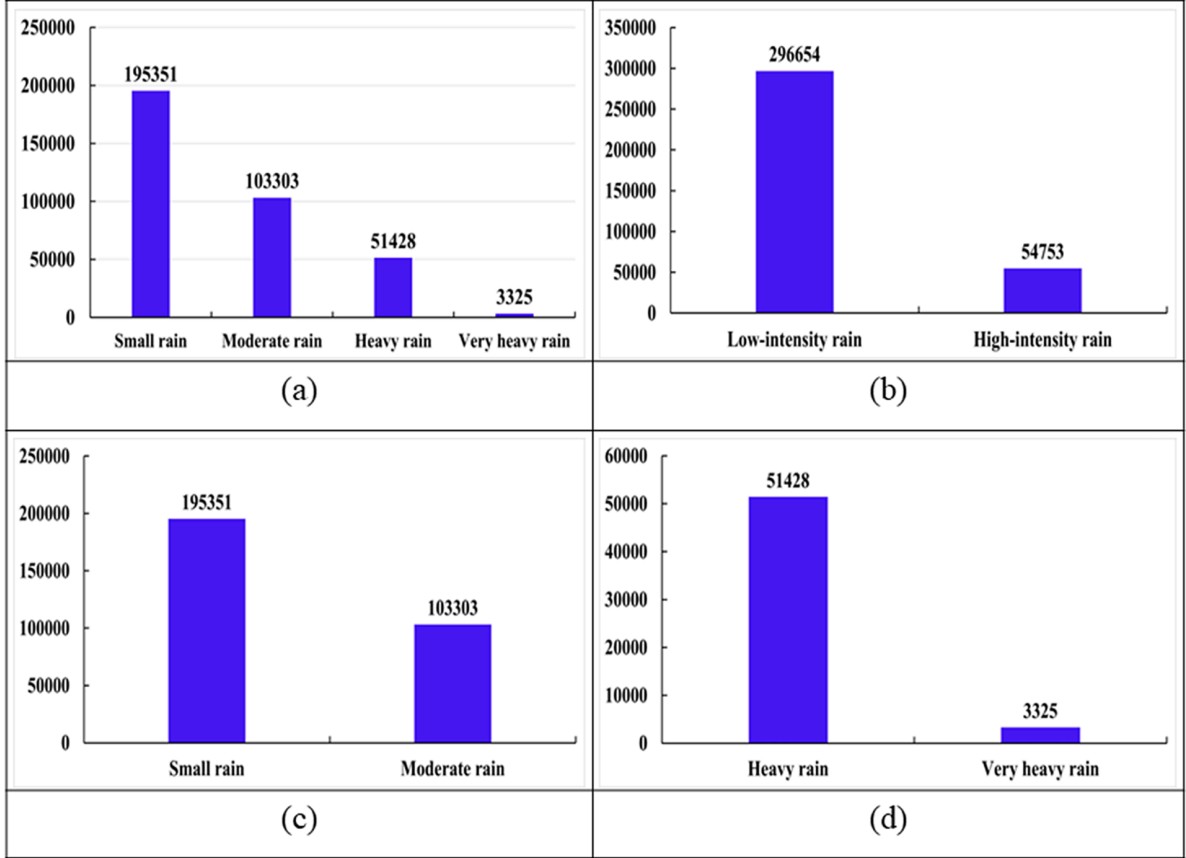

**Fig 4. Number of rainfall samples in the input datasets of the models. (a) M1, (b) M2, (c) M3, (d) M4.**

**Table 3**. Basic metrics for evaluating classification models.

| Equation | Range | Optimal |
|---|---|---|
| $CSI = \frac{TP}{TP+FP+FN}$ | $[0,1]$ | 1 |
| $POD = \frac{TP}{TP+FN}$ | $[0,1]$ | 1 |
| $ETS = \frac{TP-X}{TP+FP+FN-X}$ | $\left[-\frac{1}{3},1\right]$ | 1 |
| $With: X = \frac{(TP+FN)\times(TP+FP)}{N}$ | | |
| $HSS = 2 \times \frac{TP\times TN-FP\times FN}{(TP+FN)\times(FN+TN)+(TP+FP)\times(FP+TN)}$ | $[0,1]$ | 1 |
| $FAR = \frac{FP}{TP+FP}$ | $[0,1]$ | 0 |
| $BIAS = \frac{TP+FP}{TP+FN}$ | $[0,\infty)$ | 1 |

Similarly, the basic metrics for evaluating the performance of the regression models are presented in Table 4. Among them, the Correlation Coefficient (CC) represents the linear correlation between the estimated rainfall values from the proposed rainfall product and the actual observed rainfall. The Mean Absolute Error (MAE) and Root Mean Squared Error (RMSE) indicate the overall regression performance of the model, taking into account the correlation, bias, and error between the estimated rainfall and the actual rainfall. The Modified Kling-Gupta Efficiency (mKGE) measures variability between the estimated rainfall and the observed rainfall

**Table 4**. Basic metrics for evaluating regression models.

| Equation | Range | Optimal |
|---|---|---|
| $CC = \dfrac{\sum(p_j - \overline{p_j})(\hat{p}_i - \overline{p_i})}{\sqrt{\sum(p_j - \overline{p_j})^2}\sqrt{\sum(\hat{p}_i - \overline{p_i})^2}}$ | $[-1,1]$ | 1 |
| $MAE = \dfrac{\sum|p_i - p_j|}{N}$ | $[0, \infty)$ | 0 |
| $RMSE = \sqrt{\dfrac{\sum(p_i - p_j)^2}{N}}$ | $[0, \infty)$ | 0 |
| $mKGE = 1 - \sqrt{(CC - 1)^2 + (\beta - 1)^2 + (\gamma - 1)^2}$ | $[-\infty, 1]$ | 1 |
| $\beta = \mu_e/\mu_o \; ; \; \gamma = \sigma_e\mu_o/\sigma_o\mu_e$ | | |

Where, TP-denotes the number of rain samples correctly classified as rain (True Positives); FP-refers to the number of no-rain samples incorrectly classified as rain (False Positives); TN-represents the number of no-rain samples correctly classified as no-rain (True Negatives); FN-the number of rain samples misclassified as non-rain (False Negatives); N-the total number of samples; $p_i$, $p_j$ represent the estimated and observed rainfall values, respectively; $\overline{p_i}$, $\overline{p_j}$ denote the mean observed and estimated rainfall, respectively. The parameters $\beta$ and $\gamma$ reflect the bias and variability of the model.

## Results and discussion

This section presents and discusses the experimental results concerning feature selection, data augmentation, and hyper-parameter optimization for the training process. Additionally, the classification and regression results of each individual classification and regression model, as well as those of the proposed rainfall product, are analyzed and evaluated.

The proposed framework, investigating three algorithms (RF, XGB, LGBM) with a 3-Stage classification scheme and a cloud masking technique, was evaluated using two feature selection techniques: the Common Feature set (CF) and the RF importance-based set (RF). The CF feature selection method is applied throughout Sect 3, resulting in three corresponding rainfall products: RF-3SC-CF, XGB-3SC-CF, and LGBM-3SC-CF. The second feature selection method (RF) is only applied in Sect 3.4.4 to compare its performance against the CF method.

### Results of feature selection data augmentation and hyperparameter optimization

**Feature selection.** The selection of input features significantly impacts the quality of the proposed rainfall product. Consequently, this study presents two feature selection schemes, leading to the development of two distinct rainfall products corresponding to different input feature sets. The first strategy employs a common (fixed) feature subset, selected from the input features directly related to local weather, climate, and topography, and is applied consistently across all models from M1 to M8. The second strategy utilizes feature subsets dynamically selected by the Random Forest (RF) Importance technique, as implemented in the Scikit-learn library. This technique ranks input features based on their predictive influence, producing a prioritized list. Accordingly, a different optimal feature subset is individually selected for each model from M1 to M8 based on its specific importance ranking. The feature subsets selected to train the models are presented in Table 5.

From Table 5, it can be observed that the common feature set used to train models M1 through M8 consists of 13 features. Among these, the ERA5 features include R850, UWIND850, and VWIND850, which are selected due to their ability to reflect the development of convective clouds (Cb) in the lower troposphere, where dense air masses carrying moisture from the ocean to the mainland are more likely to induce rainfall compared to mid- and upper-level layers [39]. The TCW feature represents the total column water vapor content in the atmosphere, indicating the potential for precipitation in a given area [40]. The SLOR feature provides information on terrain slope; areas with steeper slopes (windward sides) tend to exhibit higher orographic rainfall potential [41]. The Himawari-8 BT features include four single infrared bands—WVB,

**Table 5. List of feature subsets used for model training.**

| Model | Different feature subsets selected using the RF technique | The common feature subset |
|---|---|---|
| M1 | R850, b16_irb, b11_b16, UWIND850, b14_b16, b14_i2b, b16_i2b, UWIND500, b10_irb, VWIND500, b10_b11, TCW, TCWV. | wvb, irb, i2b, i4b, wvb_irb, i2b_irb, i4b_i2b, UWIND850, VWIND850, TCW, SLOR, R850, DEM. |
| M2 | TCW, UWIND850, TCWV, R850, UWIND500, VWIND500, b12_irb, KX, b10_b14, CIN, b10_irb, b11_i2b, CAPE. | |
| M3 | UWIND850, R850, UWIND500, TCW, TCWV, KX, b16_irb, b14_b16, VWIND500, SLOR, CAPE, b16_i2b, DEM. | |
| M4 | VWIND850, VWIND250, b16_irb, UWIND850, UWIND250, R250, b11_16b, R850, i2b_irb, R500, UWIND500, TCW, KX. | |
| M5 | b12_b09, UWIND850, b14_b16, UWIND500, CAPE, VWIND500, b16_i2b, b16_irb, VWIND250, TCW, CIN, b10_b09, b14_i2b. | |
| M6 | SLOR, TCWV, UWIND850, KX, R850, UWIND500, CAPE, VWIND850, b14_b16, DEM, i2b_irb, VWIND250, IMF. | |
| M7 | CAPE, UWIND850, VWIND500, UWIND250, VWIND850, TCW, KX, VWIND250, UWIND500, R850, b10_wvb, b11_irb, b16_irb. | |
| M8 | b11_b16, VWIND250, KX, R850, UWIND250, CAPE, b11_b14, VWIND850, TCWV, VWIND500, TCW, b14_irb, R250. | |

IRB, I2B, and I4B—which are useful for identifying cloud and precipitation zones as well as surface features, and they indicate the vertical distribution of water vapor in the atmosphere. Additionally, three dual-band combinations—wvb_irb, i2b_irb, and i4b_i2b—are associated with cloud-top temperature, cloud-top height, and cloud phase. The combination of ERA5 and Himawari-8 features with the ASTER DEM feature provides comprehensive information on rainfall characteristics across regions with varying topography, weather, and climate conditions. It is important to note that the results for models M1 through M8 and the final rainfall product in Sect 3 all utilize this common dataset.

Due to the large number of input features (55 Himawari-8 features, 17 ERA5 features, and 1 ASTER DEM feature), we selected the most important features to improve training efficiency, reduce computation time, and simplify the modeling process. Additionally, to objectively compare the performance of rainfall products using features selected based on RF importance with those using a common feature set, the number of selected features used for models M1 to M8 was fixed at 13.

**Data augmentation.** The number of rain samples in different rainfall ranges, before and after applying the RR data augmentation technique with model M1, is presented in Table 6.

**Table 6. Number of samples across rainfall intervals before and after applying the RR technique.**

| Rainfall class | Value ranging (mm/h) | Rate of increase | Number of samples Before | After |
|---|---|---|---|---|
| Small | 0.1 - 1.0 | 1.0 | 195,351 | 195,351 |
| Moderate | 1.0 - 2.0 | 1.0 | 48,520 | 48,520 |
| | 2.0 - 3.5 | 1.5 | 32,439 | 48,658 |
| | 3.5 - 5.0 | 2.3 | 20,344 | 46,791 |
| Heavy | 5.0 - 8.0 | 2.4 | 20,769 | 49,845 |
| | 8.0 - 12.0 | 3.6 | 13,294 | 47,858 |
| | 12.0 - 20.0 | 4.0 | 11,944 | 47,776 |
| Very heavy | 20.0 - 30.0 | 8.0 | 5,421 | 43,368 |
| | 30.0 - 40.0 | 12.0 | 1,972 | 23,664 |
| | 40.0 - 50.0 | 25.0 | 766 | 19,150 |
| | >50.0 | 30.0 | 587 | 17,610 |

As shown in Table 6, prior to the application of the RR data augmentation technique, the M1 model exhibited a substantial imbalance in the number of samples across rainfall intervals within each rain class. Notably, this imbalance became more pronounced with increasing rainfall intensity, particularly in the heavy rain and very heavy rain categories.

To address this data imbalance issue, we increased the number of rainfall samples within each interval based on the principle that higher rainfall intensities should be associated with greater augmentation ratios. The goal was to achieve a more balanced distribution of samples across intervals within each rain class and to reduce the disparity in sample counts between different rain classes. Specifically, the small rain class, which originally had the largest number of samples, was retained without augmentation. Similarly, the 1.0–2.0 mm/h interval within the moderate rain class—due to its dominant sample count—was also kept unchanged. This interval was selected as the reference level to guide the augmentation of the remaining intervals.

For intervals with fewer samples, particularly those within the heavy and very heavy rain classes, we applied higher augmentation rates as detailed in Table 6. Our experiments indicated that increasing the augmentation rate beyond these specified values did not lead to noticeable improvements in model accuracy while significantly increasing computational complexity and training time.

**Hyperparameter optimization.** The parameters described in Table 2 were tuned using the grid search technique to determine the optimal values for each model from M1 to M8. The results of the parameter optimization process are presented in detail in Table 7.

## Cloud masks and classification models

**The CM1 and rain/no-rain classification.** This study proposes a combination of BT thresholds from Himawari-8 satellite IR bands and BT differences (BTDs), including WVB (6.2 $\mu$m), IRB (10.4 $\mu$m), I2B (12.4 $\mu$m), and the BTDs composite (2B14 $-$ I2B $-$ IRB), to identify whether a pixel in the image corresponds to rain-bearing or no-rain-bearing clouds (CM1). Additionally, the variations in BT values relative to surface rainfall measurements (Fig 5) provide the basis for selecting appropriate BT thresholds used in the construction of CM1.

As shown in Fig 5, for rainfall rates greater than 0.1 mm/h, the BT values of the observed channels tend to concentrate within the following ranges: WVB (160 K – 255 K), IRB (150 K – 300 K), I2B (160 K – 285 K), and 2B14 – I2B – IRB (0 K – 70 K). Therefore, these ranges are adopted as BT thresholds for constructing the CM1. A pixel is classified as a rain-bearing cloud if the BT values of all the aforementioned channels fall within their respective ranges. Otherwise, it is considered a no-rain-bearing cloud. By integrating CM1 with the rain/no-rain classification results from model M1, a final rain/no-rain classification map is generated. This classification map is then compared with reference radar-based rain/no-rain maps for three specific rainfall events, as illustrated in Fig 6.

Fig 6a shows the rain/no-rain classification map generated by the M1 model, where rain areas are shown in red. Fig 6b presents the cloud classification map from CM1, where areas identified as rain-bearing clouds are shown in blue. Fig 6c illustrates the rain/no-rain classification result of the proposed combined model M1 + CM1. In this approach, only the

**Table 7**. Hyperparameter optimization.

| Hyperparameter | | Optimal value | | | | | | | |
|---|---|---|---|---|---|---|---|---|---|
| *Name* | *Examined values* | *M1* | *M2* | *M3* | *M4* | *M5* | *M6* | *M7* | *M8* |
| n_estimators | 500; 700; 900; 1500; 2000; 2500 | 2000 | 2000 | 1500 | 1500 | 1500 | 900 | 1500 | 700 |
| max_depth | 8; 12; 14; 16; 18; 24; 28; 32, 36 | 28 | 28 | 24 | 24 | 24 | 18 | 18 | 16 |
| subsample | 0.6; 0.8; 1.0 | 0.8 | 0.8 | 0.8 | 0.8 | 0.8 | 0.8 | 0.8 | 0.8 |
| colsample_bytree | 0.6; 0.8; 1.0 | 0.8 | 0.8 | 0.8 | 0.8 | 0.8 | 0.8 | 0.8 | 0.8 |
| Class weight | 1.6; 1.8; 2.0; 2.5; 3.0; 5.5; 7.0; 9.5; 10.5 | | 1.8 | 1.6 | 9.5 | | | | |
| learning_rate | 0.1; 0.01; 0.001; 0.002; 0.005, 0.08 | 0.01 | 0.01 | 0.01 | 0.01 | 0.005 | 0.005 | 0.002 | 0.002 |

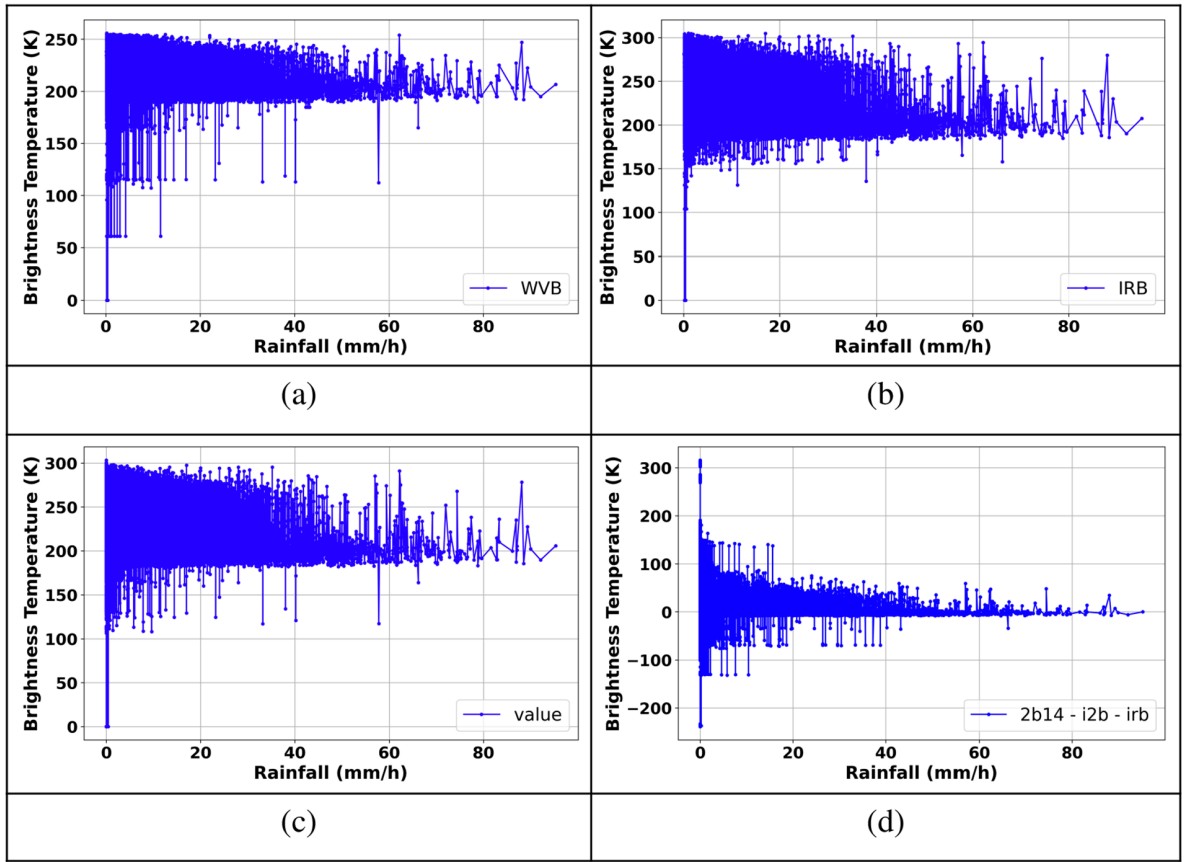

**Fig 5. Variation of BT values of IR bands with respect to rainfall.** (a) WVB, (b) IRB, (c) I2B, (d) 2B14-I2B-IRB.

locations classified as rain by M1 and simultaneously identified as rain-bearing clouds by CM1 are labeled as rain (red); all other pixels are classified as no-rain.

For the rainfall event at 00:00 on October 6, 2020, the rain/no-rain classification map produced by the proposed M1 + CM1 model (Fig 6c) shows a higher correlation with the radar-based classification than the original M1 model (Fig 6a). In Fig 6c, the area classified as rain (red) is smaller than in Fig 6a. Notably, the northern and northeastern parts of the map, which were classified as rain by the M1 model (Fig 6a) but not identified as rain-bearing clouds by CM1 (Fig 6b), were excluded. As a result, the rain areas (red) in Fig 6c are more spatially constrained compared to Fig 6a.

In contrast, for the rainfall events at 06:00 on November 28, 2011, and 12:00 on September 2, 2020, the rain areas identified by the M1 model (Fig 6a) are entirely within the areas classified as rain-bearing clouds by CM1 (Fig 6b). There-fore, after removing cloud regions not associated with rain in M1, the rain/no-rain classification map of the proposed M1 + CM1 model (Fig 6c) remains effectively identical to the original M1 model (Fig 6a).

Overall, across the three evaluated rainfall events, the three proposed models, including LGBM, XGB, and RF, demon-strated high performance in classifying rain/no-rain areas. Among them, the combined rain/no-rain classification with the cloud-mask integration (M1 + CM1) produced by the LGBM model shows a higher level of agreement with the radar-based rainfall maps compared to those generated by the XGB and RF models.

**The CM2 and low/high-intensity rainfall classification.** To identify regions of clouds bearing high-intensity rainfall, this study proposes a combination of BT thresholds from four IR channels of the Himawari-8 satellite, including WVB, I4B

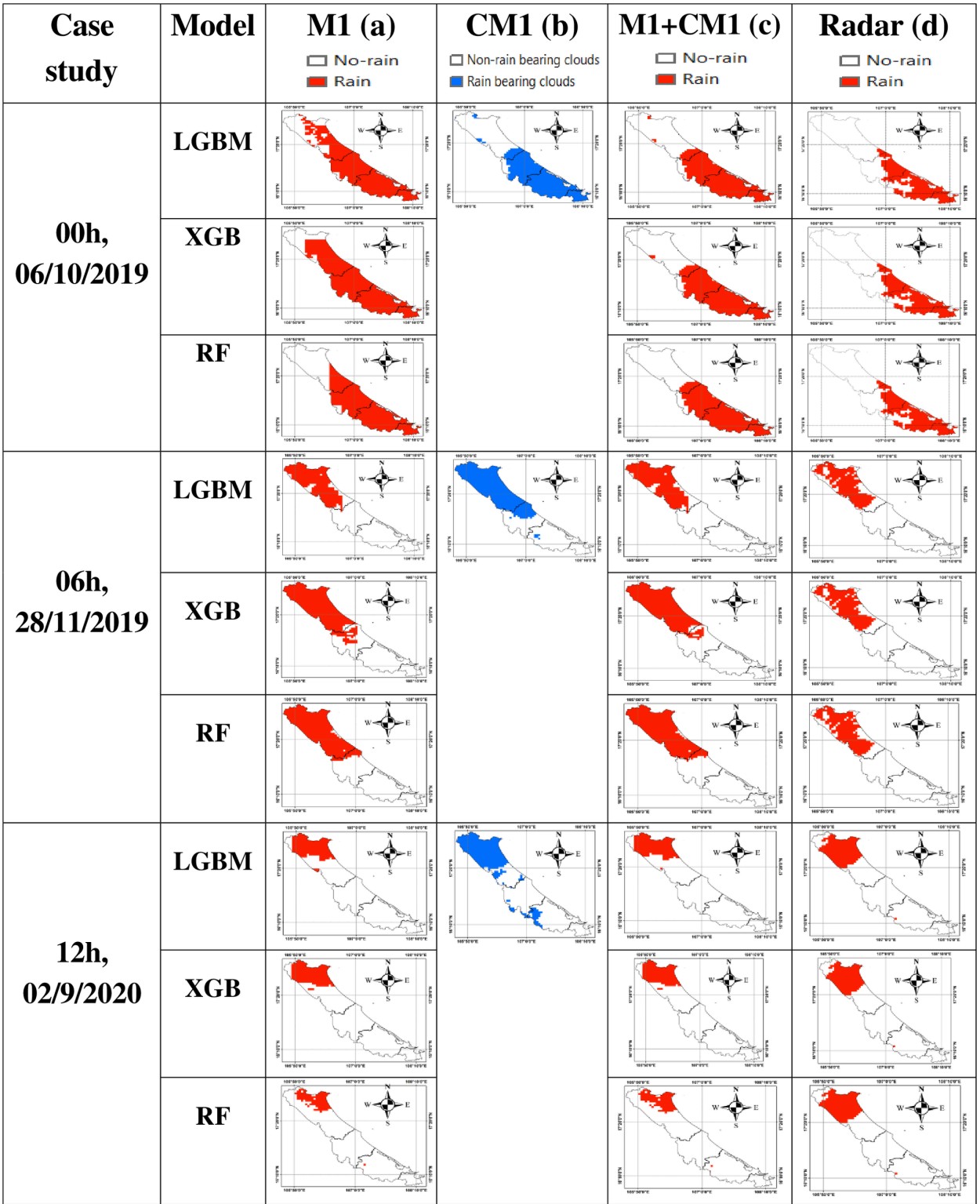

**Fig 6**. **Rain/no-rain classification maps from the proposed model and radar.** (a) rain/no-rain classification maps from model M1; (b) classification maps of rain/no-rain-bearing clouds based on CM1; (c) combined rain/no-rain classification maps from M1+CM1; (d) reference rain/no-rain classification maps from radar observations.

(3.9 $\mu m$), IRB, and B14 (11.2 $\mu m$). The classification map that distinguishes between clouds bearing low-intensity rain and clouds bearing high-intensity rain is referred to as CM2. The physical characteristics of these channels are described in Sect 2.3.2, along with the variations in BT values corresponding to surface rainfall rates exceeding 5.0 mm/h, obtained from ground-based rain gauge stations (Fig 7). These provide the basis for selecting appropriate BT thresholds for the development of CM2.

As shown in Fig 7, for rainfall rates greater than 5.0 mm/h, the BT values of the IR channels tend to fall within the following ranges: WVB (185 K – 240 K), IRB (185 K – 260 K), I4B (180 K – 270 K), and B14 (180 K – 240 K). Accordingly, these BT ranges are adopted as thresholds for constructing CM2. A location is identified as clouds bearing high-intensity rainfall if the BT values of the above IR channels satisfy the selected threshold conditions. The high-intensity/low-intensity rain classification maps produced by the proposed model, which integrates M2 and CM2 (denoted as M2 + CM2), were generated for three specific rainfall events and compared with the corresponding radar-based classification maps, as detailed in Fig 8.

Fig 8a illustrates the classification results of the M2 model for low-intensity and high-intensity rainfall, where areas classified as high-intensity rain are shown in pink. Fig 8b presents the cloud classification results from the CM2 model, in which regions identified as high-intensity rain-bearing clouds are marked in red. Fig 8c shows the classification results of the proposed M2 + CM2 combined model for high-intensity/low-intensity rain. In this approach, only the locations classified as high-intensity rainfall by M2 (pink) and simultaneously identified by CM2 as high-intensity rain-bearing clouds (red) are labeled as high-intensity rainfall (pink); all remaining pixels are labeled as low-intensity rainfall (light yellow).

For the rainfall event at 00:00 on October 6, 2020, the high-intensity/low-intensity rain classification map generated by the proposed M2 + CM2 model (Fig 8c) shows a higher level of consistency with the radar-based rain classification map

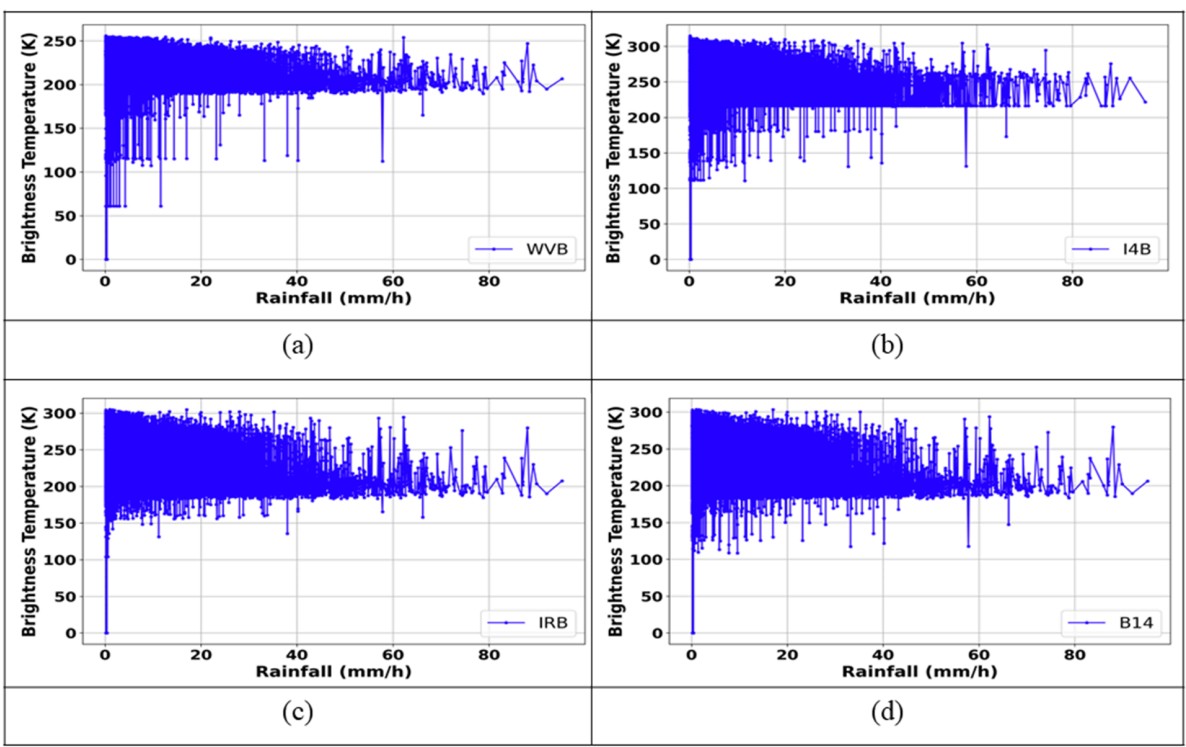

**Fig 7**. **Variation of BT values from IR bands for rainfall rates** > 5.0 mm/h. (a) WVB, (b) I4B, (c) IRB, (d) B14.

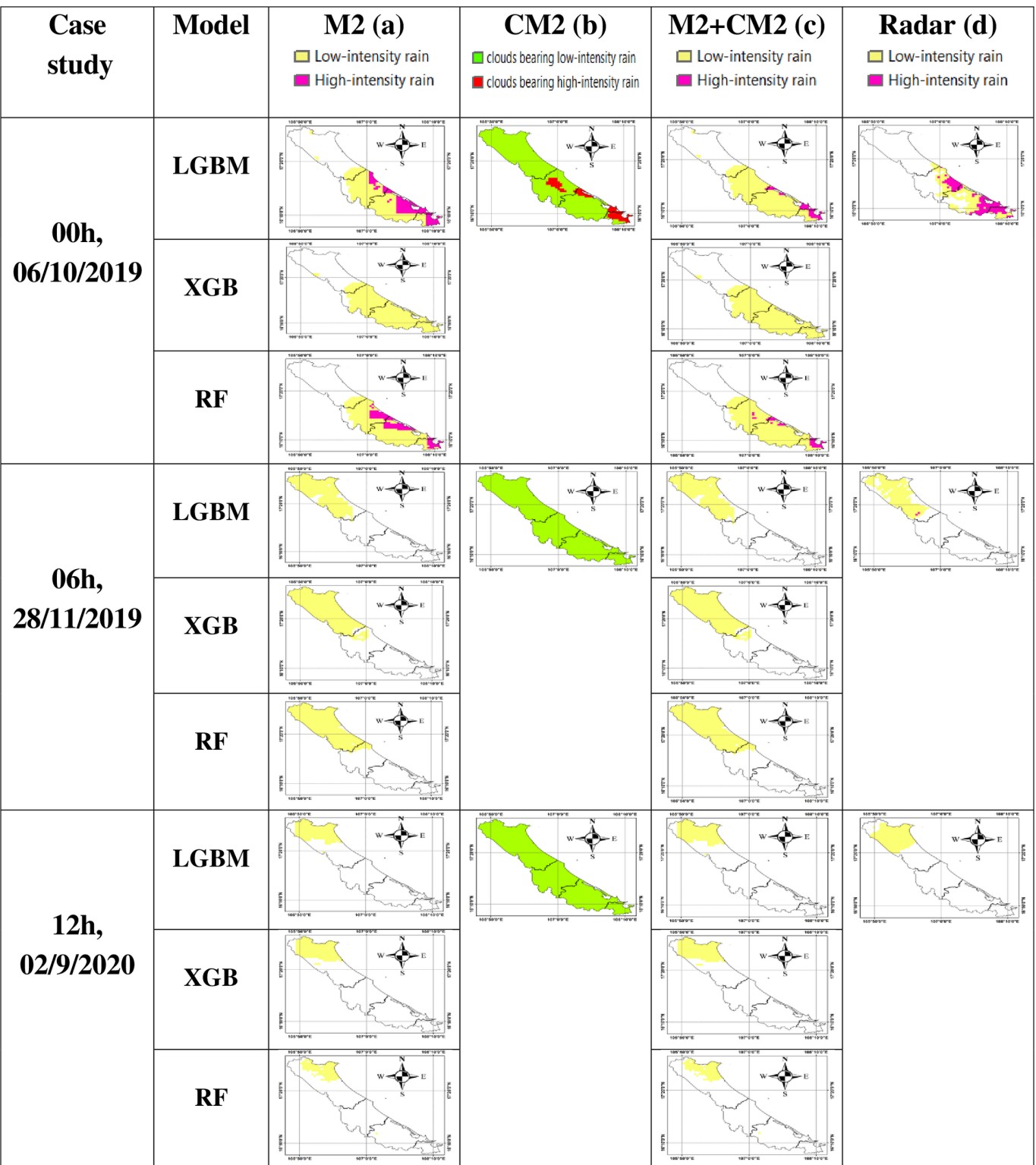

**Fig 8**. **Classification maps of low-intensity rain and high-intensity rain from the proposed model and radar.** (a) Classification of low-intensity rain and high-intensity rain from model M2; (b) Classification of clouds bearing low-intensity and clouds bearing high-intensity rain based on CM2; (c) Combined classification results from M2 + CM2; (d) Radar-based classification of low-intensity rain and high-intensity rain.

(Fig 8d) than the classification map produced by the M2 model alone (Fig 8a), which does not incorporate cloud masking. Specifically, areas that were classified as high-intensity rain by M2 but not identified as high-intensity rain-bearing clouds by CM2 were excluded, resulting in a more accurate and spatially compact distribution of high-intensity rainfall (pink) in Fig 8c compared to Fig 8a.

In contrast, for the rainfall events at 06:00 on November 28, 2019, and 12:00 on September 2, 2020, the high-intensity/low-intensity rain classification maps generated by the proposed M2 + CM2 model (Fig 8c) appear similar to both the M2-only classification results and the radar-based classification maps. This can be explained by the fact that CM2 classified most of the cloud regions in these cases as low-intensity rain-bearing clouds. As a result, the regions initially labeled by M2 as low-intensity rain remained unchanged after combining with CM2, leading to no significant difference in the final classification maps.

From Fig 8, it can also be observed that, overall, the low-intensity and high-intensity rainfall classification maps produced by the three evaluated models exhibit a high level of consistency with the radar classification maps. Among them, the LGBM model shows the highest degree of agreement. However, the classification of high-intensity rainfall areas remains less accurate, which is closely related to the inherently complex variability of rainfall intensity.

**The detailed rainfall classification models M3 and M4.** After the low- and high-intensity rain areas are accurately delineated based on the M2 + CM2 combination, these regions are subsequently input into Models M3 and M4 for further classification into four rainfall intensity categories: small rain, moderate rain, heavy rain, and very heavy rain. To visually assess the rainfall classification performance of the proposed M2+CM2 product, three representative rainfall classification maps were selected for evaluation and are presented in Fig 9.

In Fig 9, the final rainfall classification maps produced by the proposed LGBM, XGB, and RF models all exhibit a high degree of consistency with the radar-based classification map. Among them, the LGBM model provides the highest level of agreement, followed by the XGB and RF models. The regions corresponding to small and moderate rainfall show the highest agreement, while the correlation gradually decreases for the heavy rainfall class. The area classified as heavy rainfall by the proposed model appears more spatially widespread compared to that identified by radar. This discrepancy may be attributed to the localized and short-lived nature of heavy rainfall events, which often occur over narrow spatial and temporal scales and are typically associated with complex weather conditions such as strong winds and thunderstorms, posing significant challenges for accurate observation and classification.

## Performance evaluation of proposed rainfall products

**Classification performance of the proposed rainfall product.** The proposed rainfall products generated using the LGBM, XGB, and RF algorithms are referred to as LGBM-3SC-CF, XGB-3SC-CF, and RF-3SC-CF, respectively. All of these products incorporate both data balancing techniques and cloud masking. Their rainfall classification performance is compared with versions that do not use cloud masking (denoted as LGBM-3S, XGB-3S, and RF-3S) as well as with four regional satellite-based rainfall products using a dataset of 7,851 rainfall maps. These regional products include IMERG Final Run, IMERG Early Run, GSMaP_MVK_Gauge, and PERSIANN_CCS. All products are resampled to a uniform spatial and temporal resolution prior to evaluation based on observations from rain gauge stations (Fig 10).

In Fig 10, the classification performance of the rainfall products is evaluated using four metrics: POD, CSI, $1 - FAR$, and BIAS. The proposed rainfall products—LGBM-3SC-CF, XGB-3SC-CF, and RF-3SC-CF—all demonstrate strong rainfall classification performance. Among them, LGBM-3SC-CF achieves the highest $1 - FAR$ value of 0.70, representing improvements of 7.70% and 11.11% over XGB-3SC-CF and RF-3SC-CF, respectively. Regarding BIAS, LGBM-3SC-CF obtains a value of 1.04, which is the best among the three proposed products, improving upon XGB-3SC-CF and RF-3SC-CF by 14.05% and 15.45%, respectively. In terms of CSI, both LGBM-3SC-CF and XGB-3SC-CF achieve the highest result of 0.55, while RF-3SC-CF performs slightly lower at 0.53. For POD, XGB-3SC-CF, RF-3SC-CF, and LGBM-3SC-CF

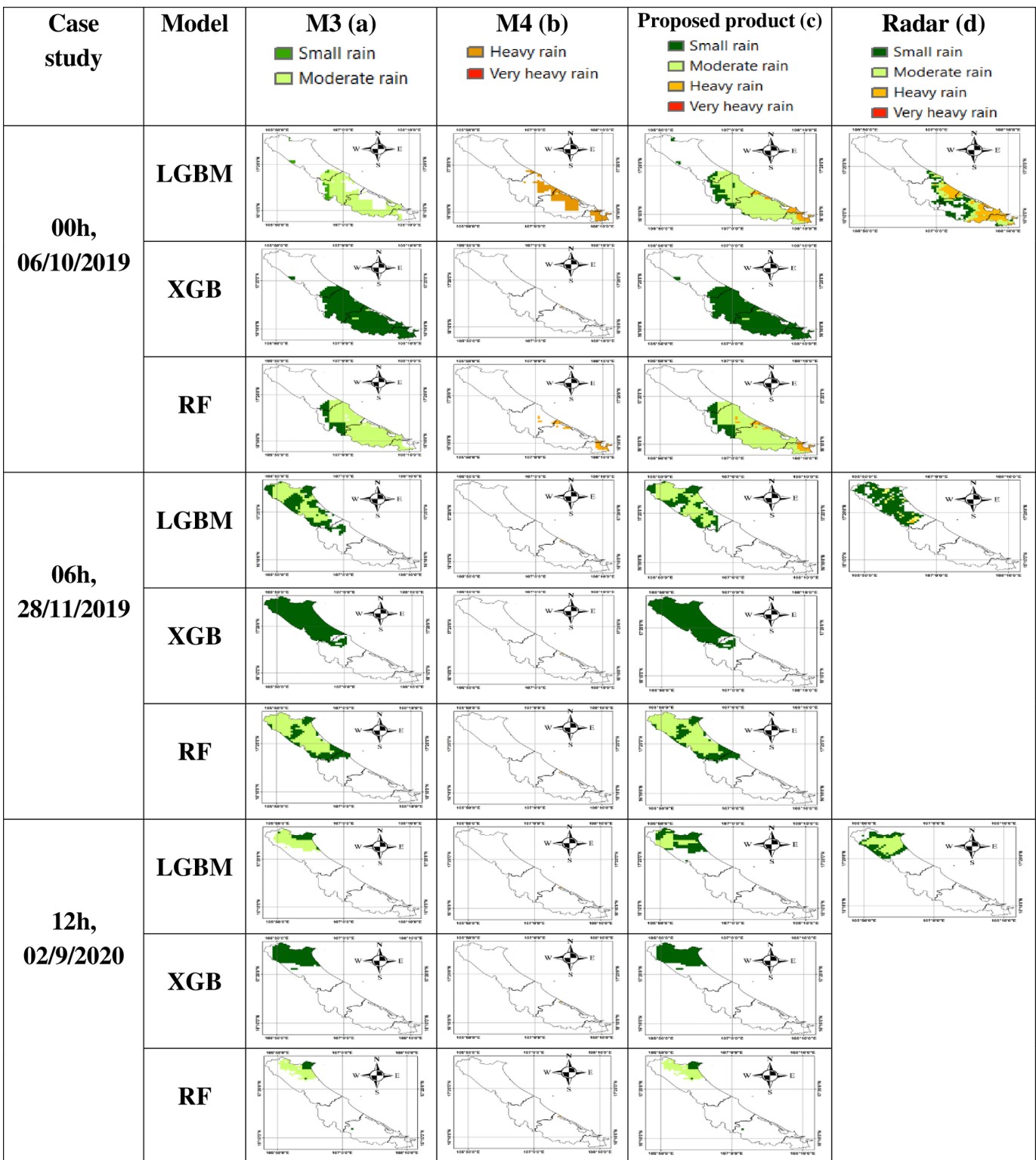

**Fig 9**. **Rainfall classification maps of the proposed product and radar.** (a) Classification of small rain and moderate rain by model M3, applied to areas identified as low-intensity rain; (b) Classification of heavy rain and very heavy rain by model M4, applied to areas identified as high-intensity rain; (c) The final rainfall classification map of the proposed rainfall product; (d) Reference rainfall classification map from radar.

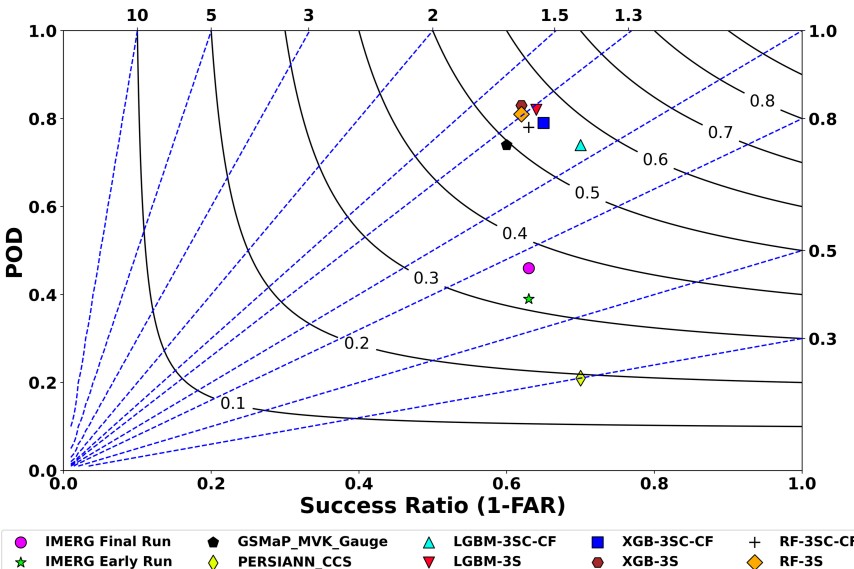

**Fig 10. Classification performance of rainfall products.** The black solid line represents the CSI, while the blue dashed line indicates the BIAS.

attain values of 0.79, 0.78, and 0.74, respectively. Overall, these results indicate that LGBM-3SC-CF delivers the highest rainfall classification performance among the three proposed products.

When comparing the proposed LGBM-3SC-CF with its variant that does not incorporate cloud masking (LGBM-3S), LGBM-3SC-CF exhibits better performance in terms of $1 - \text{FAR}$ and BIAS. Specifically, $1 - \text{FAR}$ increases by 9.38%, and BIAS improves by 18.75%. However, it is noteworthy that LGBM-3SC-CF shows a 1.82% lower CSI and a 10.81% lower POD compared with LGBM-3S. Compared with the delayed products, LGBM-3SC-CF outperforms the best-performing product in this group, GSMaP_MVK_Gauge, with improvements of 14.58% in CSI, 16.67% in $1 - \text{FAR}$, and 16.13% in BIAS, while the POD values of both products are identical at 0.74. Among the near-real-time products, IMERG Early Run shows the best performance in terms of CSI, POD, and BIAS. However, LGBM-3SC-CF still demonstrates superior performance, with improvements of 71.88% in CSI, 89.74% in POD, and 89.47% in BIAS. Across all evaluated rainfall products, PERSIANN_CCS and LGBM-3SC-CF exhibit the highest $1 - \text{FAR}$ value, both reaching 0.70.

In addition, this study compares the rainfall classification performance of the proposed product LGBM-3SC-CF with radar imagery based on rain gauge station data using several evaluation metrics, including POD, CSI, ETS, FAR, and BIAS. The detailed comparison results are presented in Table 8.

Table 8 indicates that the proposed LGBM-3SC-CF product outperforms radar in terms of POD, CSI, ETS, and BIAS by 37.04%, 12.24%, 7.50%, and 93.65%, respectively. However, the $1 - \text{FAR}$ of the radar (0.85) is 21.43% higher than that of the proposed LGBM-3SC-CF product. This implies that the LGBM-3SC-CF tends to classify more no-rain locations as rain compared to the radar.

**Table 8. Rain classification performance of the proposed product and radar.**

| Rainfall product | POD | CSI | ETS | FAR | BIAS |
|---|---|---|---|---|---|
| LGBM-3SC-CF | **0.74** | **0.55** | **0.43** | 0.30 | **1.04** |
| Radar | 0.54 | 0.49 | 0.40 | **0.15** | 0.63 |

**Classification performance evaluation in representative case studies.** To visually assess the rain classification performance of the proposed LGBM-3SC-CF product and regional rainfall products, rain/no-rain classification maps from these products were compared against the radar-based classification map, which serves as the reference. Four randomly selected rainfall events, representing different time points among a total of 7851 rain/no-rain classification maps, were chosen for comparison and are illustrated in Fig 11.

It is evident that for the first four rainfall events, the proposed LGBM-3SC-CF product successfully detected 100% of the rainfall pixels identified by the radar-based benchmark. Similarly, during the final event (21:00 on June 5, 2021), the model accurately detected rainfall points in the west-central region of the map. Furthermore, the area misclassified as rain (false alarm) by LGBM-3SC-CF was consistently smaller than that of the best-performing gauge-corrected product, GSMaP_MVK_Gauge, across most classification maps.

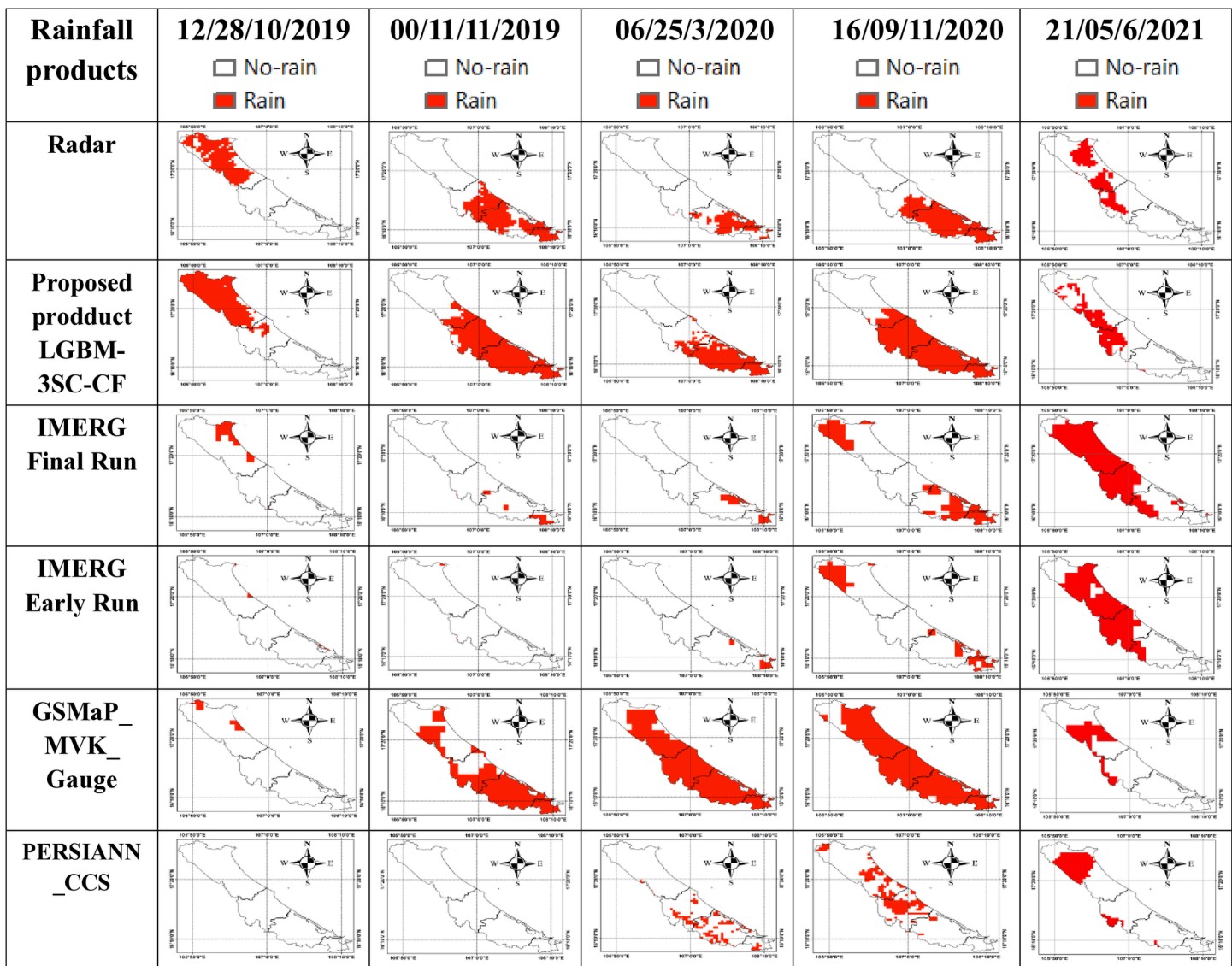

**Fig 11**. Rain/no-rain classification maps of the rainfall products.

In contrast, the GSMaP_MVK_Gauge product missed a significant number of actual rainfall pixels—indicated by a smaller red area compared to the radar data—during the events at 12:00 on October 28, 2020, and 21:00 on June 5, 2021. Conversely, it exhibited substantial overestimation by misclassifying numerous non-rain areas as rain in the remaining three events.

Additionally, this study conducted a quantitative comparison of the rain/non-rain classification performance of the proposed LGBM-3SC-CF product and the regional rainfall products based on their ability to accurately detect rain points, using the CSI and POD metrics, as presented in Table 9.

Table 9 indicates that the proposed rainfall product, LGBM-3SC-CF, achieved the highest classification performance across three specific rainfall events: 12:00 on October 28, 2019; 00:00 on November 11, 2019; and 16:00 on November 9, 2020. For the event at 12:00 on October 28, 2019, LGBM-3SC-CF attained a CSI of 0.40 and a POD of 0.61, representing improvements of 37.93% and 32.61%, respectively, over the best delayed product (IMERG Final Run). When compared to the best real-time product (IMERG Early Run), LGBM-3SC-CF exhibited even more substantial improvements of 81.82% in CSI and 60.53% in POD.

Regarding the event at 00:00 on November 11, 2019, LGBM-3SC-CF achieved a CSI of 0.22 and a POD of 0.68. This corresponds to an increase of 4.55% and 25.93%, respectively, compared to the best gauge-calibrated product (GSMaP_MVK_Gauge). Against the best real-time product, the proposed model demonstrated remarkable improvements of 1000% in CSI and 871.43% in POD. In the event at 16:00 on November 9, 2020, LGBM-3SC-CF yielded a CSI of 0.42, which was 13.51% higher than that of the best delayed product (IMERG Final Run). Its POD stood at 0.89, outperforming the best gauge-calibrated product (GSMaP_MVK_Gauge) by 3.49%.

Conversely, for the event at 21:00 on June 5, 2021, the LGBM-3SC-CF product recorded a CSI of 0.27, which was comparable to the best real-time product (PERSIANN-CCS) and 12.5% higher than the best delayed product (IMERG Final Run). However, its POD of 0.64 was lower than that of IMERG Final Run, GSMaP_MVK_Gauge, and IMERG Early Run, though it remained superior to PERSIANN-CCS. Notably, in the rainfall event at 06:00 on March 25, 2020, the proposed LGBM-3SC-CF yielded a CSI of 0.38, which was 13.16% lower than IMERG Final Run and 7.32% lower than IMERG Early Run, but still 40.74% higher than GSMaP_MVK_Gauge. For POD, LGBM-3SC-CF achieved a value of 0.86, which was 12.17% lower than GSMaP_MVK_Gauge, yet 45.76% and 72.0% higher than IMERG Final Run and IMERG Early Run, respectively.

Overall, the proposed LGBM-3SC-CF product demonstrated the highest rainfall classification performance among the evaluated products.

**Regression performance of the proposed rainfall product.** To evaluate the regression performance of the proposed rainfall products, including LGBM-3SC-CF, XGB-3SC-CF, and RF-3SC-CF, this study compares them with regional

**Table 9**. Classification performance of rainfall products compared to radar in five rainfall events.

| Case study | | Rainfall products | | | | |
|---|---|---|---|---|---|---|
| | | IMERG Final Run | IMERG Early Run | GSMaP_MVK_Gauge | PERSIANN_CCS | Proposed product LGBM-3SC-CF |
| 12h00 28/10/2019 | CSI | 0.29 | 0.22 | 0.09 | 0.02 | **0.4** |
| | POD | 0.46 | 0.38 | 0.11 | 0.02 | **0.61** |
| 00h00 11/11/2019 | CSI | 0.07 | 0.02 | 0.21 | 0 | **0.22** |
| | POD | 0.3 | 0.07 | 0.54 | 0 | **0.68** |
| 06h00 25/3/2020 | CSI | **0.43** | 0.41 | 0.24 | 0.12 | 0.38 |
| | POD | 0.59 | 0.5 | **0.97** | 0.19 | 0.86 |
| 16h00 9/11/2020 | CSI | 0.37 | 0.39 | 0.31 | 0.01 | **0.42** |
| | POD | 0.8 | 0.65 | 0.86 | 0.02 | **0.89** |
| 21h00 5/6/2021 | CSI | 0.24 | 0.21 | 0.17 | 0.27 | **0.27** |
| | POD | 0.71 | **0.73** | 0.72 | 0.42 | 0.64 |

rainfall products using ground-based observations from rain gauge stations. The evaluation is conducted using several metrics, including CC, MAE, RMSE, and mKGE, and the detailed results are presented in Fig 12.

Fig 12 shows that all cloud mask combined rainfall products, including LGBM-3SC-CF, XGB-3SC-CF, and RF-3SC-CF, achieve higher rainfall regression performance compared to their counterparts that do not combine a cloud mask. Specifically, in terms of the mKGE metric, LGBM-3SC-CF outperforms LGBM-3S by 6.82%, XGB-3SC-CF exceeds XGB-3S by 2.27%, and RF-3SC-CF outperforms RF-3S by 10.0%. For the error metrics MAE and RMSE, LGBM-3SC-CF shows improvements over LGBM-3S of 6.01% and 2.84%, respectively; XGB-3SC-CF achieves improvements of 0.96% and 0.63% over XGB-3S; and RF-3SC-CF improves upon RF-3S by 1.72% and 0.96%, respectively. Regarding the CC metric, LGBM-3SC-CF, LGBM-3S, and XGB-3SC-CF achieve the highest value of 0.47, followed closely by XGB-3S at 0.46, while both RF-3SC-CF and RF-3S yield a CC of 0.44. Thus, LGBM-3SC-CF is the proposed rainfall product with the best regression performance, achieving a CC of 0.47, an mKGE of 0.47, an MAE of 2.66 mm/h, and an RMSE of 5.48 mm/h.

Compared with the delayed rainfall products, LGBM-3SC-CF shows improvements over GSMaP_MVK_Gauge and IMERG Final Run in terms of CC, with improvements of 6.82% over both products. For the mKGE metric, LGBM-3SC-CF outperforms the best delayed rainfall product, IMERG Final Run, by 17.5%. Regarding RMSE, LGBM-3SC-CF achieves a 0.72% improvement over GSMaP_MVK_Gauge, which is the best-performing delayed product. In contrast, for MAE, LGBM-3SC-CF performs slightly worse than GSMaP_MVK_Gauge by 6.39%, but it still ranks as the second-best among

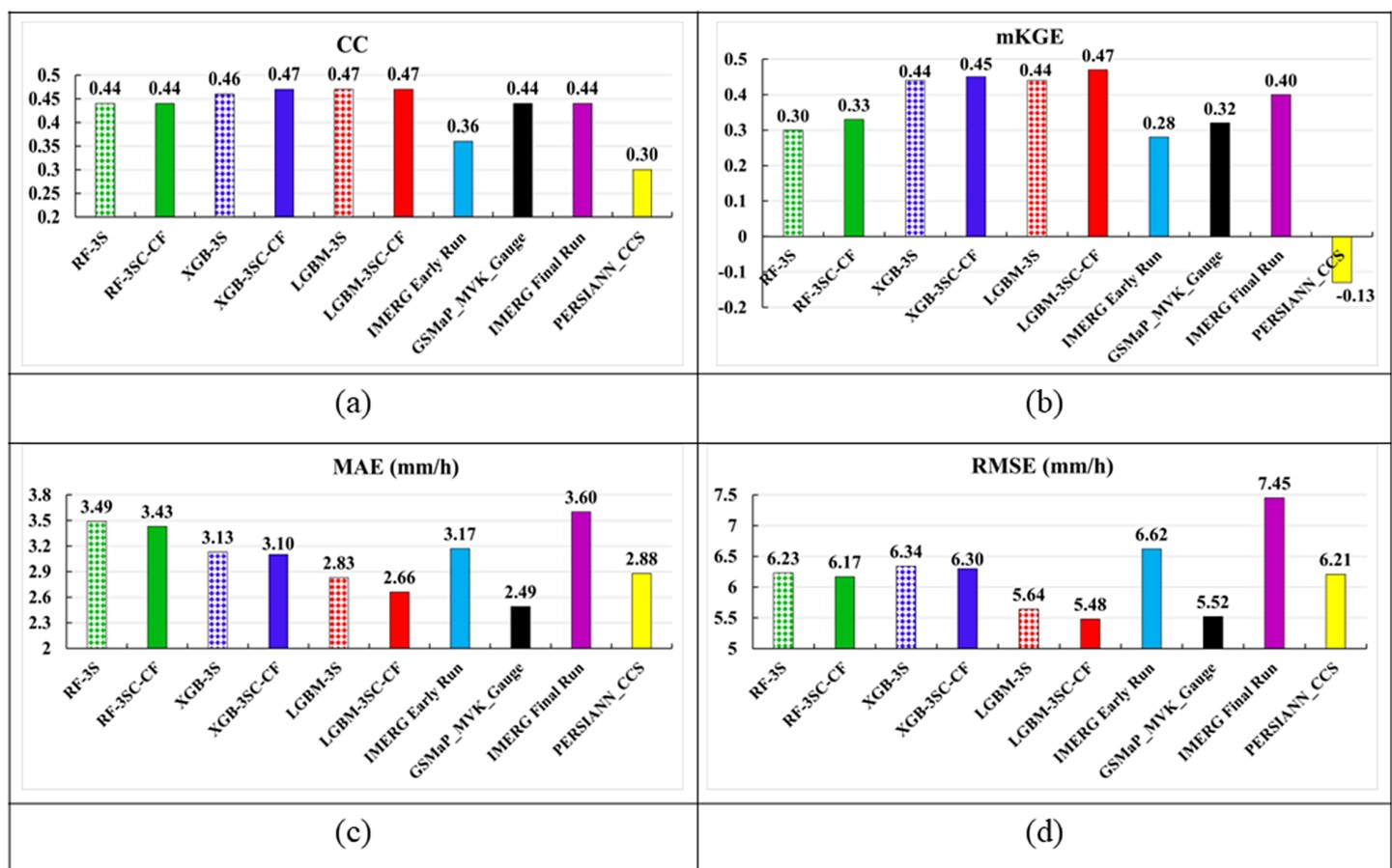

**Fig 12. Regression performance of rainfall estimation: (a) CC, (b) mKGE, (c) MAE, (d) RMSE.**

all evaluated rainfall products. Similarly, when compared with near-real-time rainfall products, LGBM-3SC-CF also outperforms the best-performing product, PERSIANN_CCS, with improvements of 7.64% in MAE and 11.57% in RMSE. In terms of CC and mKGE, LGBM-3SC-CF exhibits superior performance relative to the best near-real-time product, IMERG Early Run, with improvements of 30.55% and 67.86%, respectively.

In addition, this study evaluates the rainfall regression performance of the proposed LGBM-3SC-CF product against radar observations at rain gauge stations in the study area, using the mKGE and BIAS metrics. The detailed comparison results are presented in Table 10.

The results in Table 10 indicate that the proposed rainfall product, LGBM-3SC-CF, demonstrates superior regression performance compared to radar, with an improvement of 2.17% in mKGE and 86.15% in BIAS.

**Estimation performance evaluation in representative case studies.** To visually evaluate the rainfall estimation performance of the proposed LGBM-3SC-CF product and regional rainfall products, rainfall maps from these products were compared against reference radar-based rainfall maps. Four rainfall events previously selected for classification performance assessment in Sect 3.3.3 were further examined and are presented in detail in Fig 13.

In Fig 13, rainfall intensity across the region is represented using different colors: non-rain areas are shown in white, while areas with rainfall greater than 0.1 mm/h are displayed from dark blue to red (rainfall intensities exceeding 30.0 mm/h), reflecting the magnitude of rainfall at each location. It is evident that the rainfall maps produced by the proposed LGBM-3SC-CF product closely resemble the reference radar-based rainfall maps. Specifically, for the rainfall event at 12:00 on October 28, 2019, rainfall is concentrated in the northern and northwestern areas of the map. For the events at 00:00 on November 11, 2019; 06:00 on March 25, 2020; and 16:00 on November 9, 2020, rainfall is primarily concentrated in the southern and southeastern regions. During the rainfall event at 21:00 on June 5, 2021, rainfall values were distributed mainly in the west-central part of the map. In contrast, the regional rainfall products exhibit substantial deviations in both rainfall coverage and intensity, often failing to match the locations indicated by the radar data.

In addition, this study quantitatively evaluates the rainfall regression performance of the proposed LGBM-3SC-CF product and the regional rainfall products for the four selected events using the RMSE metric, with the results presented in Table 11.

Table 11 shows that, across all five rainfall events examined, the proposed LGBM-3SC-CF rainfall product achieves the lowest RMSE among all evaluated rainfall products. Specifically, the RMSE of LGBM-3SC-CF demonstrates improvements over the best-performing comparative products as follows: a 7.72% reduction compared to IMERG Final Run for the event at 12:00 on 18 October 2019; an 87.42% reduction at 00:00 on 11 November 2019; a 14.65% reduction compared to IMERG Early Run at 16:00 on 9 November 2020; an 87.69% reduction compared to GSMaP_MVK_Gauge for the event at 06:00 on 23 March 2020; and a 2.80% improvement for the rainfall event at 21:00 on 5 June 2021.

### Further discussion

**Impact of multi-stage classification.** The performance of the proposed LGBM-3SC-CF product, which employs an architecture with three rainfall classification stages, was compared with rainfall products LGBM-2S and LGBM-1S, which utilize the same input features but incorporate two and one classification stages, respectively. The comparison was based on four classification metrics and two regression metrics (mKGE and BIAS), as presented in Table 12.

In terms of classification performance, Table 12 indicates that all three rainfall products exhibit comparable CSI and ETS indices. However, regarding FAR and Bias, the proposed LGBM-3SC-CF product achieves the best classification

**Table 10. Regression performance comparison with radar.**

| Rainfall product | mKGE | BIAS |
| --- | --- | --- |
| LGBM-3SC-CF | **0.47** | **1.09** |
| Radar | 0.46 | 0.65 |

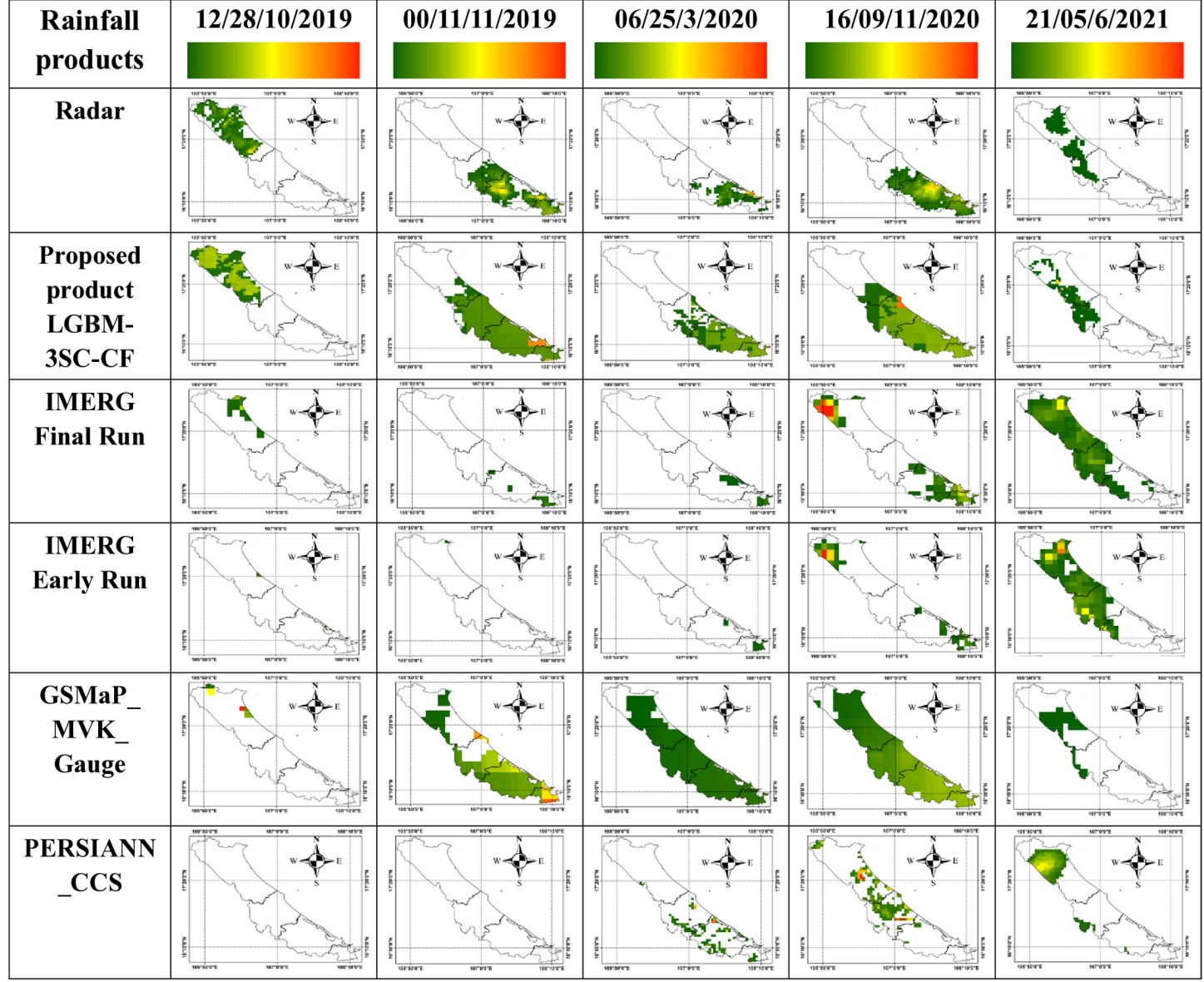

**Fig 13**. Detailed rainfall maps of the rainfall products.

**Table 11**. RMSE values of the different rainfall products compared with radar in the five rainfall events.

| Case study | Rainfall products | | | | |
|---|---|---|---|---|---|
| | IMERG Final Run | IMERG Early Run | GSMaP_ MVK_Gauge | PERSIANN_ CCS | Proposed product LGBM-3SC-CF |
| 12h00, 28/10/2019 | 2.61 | 2.46 | 3.04 | 2.87 | **2.27** |
| 00h00, 11/11/2019 | 3.18 | 3.26 | 3.83 | 3.27 | **0.40** |
| 06h00, 25/3/2020 | 6.49 | 6.54 | 5.77 | 6.93 | **0.71** |
| 16h00, 09/11/2020 | 3.56 | 3.55 | 3.81 | 4.51 | **3.03** |
| 21h00, 05/6/2021 | 5.80 | 6.69 | 4.64 | 4.75 | **4.51** |

**Table 12**. Rainfall regression performance of the rainfall products using the different classification architectures.

| Rainfall products | Classification metrics | | | | Regression metrics | |
|---|---|---|---|---|---|---|
| | CSI | ETS | FAR | BIAS | mKGE | BIAS |
| LGBM-3SC-CF | 0.55 | **0.43** | **0.30** | **1.04** | **0.46** | **1.04** |
| LGBM-2S | 0.55 | 0.43 | 0.31 | 1.06 | 0.18 | 1.44 |
| LGBM-1S | **0.56** | 0.43 | 0.34 | 1.17 | 0.23 | 1.34 |

performance, with a FAR of 0.30 and a Bias of 1.04. These results outperform the second-best product, LGBM-2S, by 3.23% and 33.33%, and the third-best product, LGBM-1S, by 11.76% and 76.47%, respectively.

Similarly, the proposed LGBM-3SC-CF product demonstrates superior rainfall regression performance compared to the LGBM-2S and LGBM-1S products. Specifically, LGBM-3SC-CF achieves an mKGE of 0.47, which is 49.0 times higher than that of LGBM-2S and 2.47 times higher than that of LGBM-1S. In terms of the BIAS metric, LGBM-3SC-CF attains a value of 1.09, representing an improvement of 1.43 times compared to LGBM-2S and 1.26 times compared to LGBM-1S.

**Impact of multi-class classification.** To evaluate the efficiency of the proposed framework in classifying rainfall into four detailed categories (small, moderate, heavy, and very heavy), this study compares it with a similar framework presented in [5]. In [5], rainfall was classified into two classes, including light and heavy rain, based on a rainfall threshold of 1.8 mm/h. Specifically, the proposed LGBM-3SC-CF product is compared against the product based on the framework from [5] that utilizes the same cloud masks (CM1 and CM2) and feature set as LGBM-3SC-CF. The rainfall regression performance of these two products was evaluated using metrics CC, MAE, RMSE, mKGE, and BIAS, with detailed results presented in Table 13.

As shown in Table 13, the proposed rainfall LGBM-3SC-CF product demonstrated superior rainfall regression performance compared to LGBM-2SC-CF (which employs the framework referenced from Hirose et al., 2019 [5]). Specifically, it achieved improvements of 6.82% in CC, 9.83% in MAE, 5.29% in RMSE, 14.63% in mKGE, and 47.06% in BIAS.

**Impacts of using data balancing techniques on rainfall classification.** A comparison of the F1-scores on the test set for the minority classes of the rainfall classification models—M1 (rain class), M2 (high-intensity rain), M3 (moderate rain), and M4 (very heavy rain)—under both balanced and imbalanced data scenarios is presented in detail in Fig 14. As shown in Fig 14, the classification performance on the minority class for all rainfall classification models (M1–M4) significantly improved after balanced data compared to the imbalanced data case.

Specifically, for model M1, the F1-score for the rain class after data balancing improved by 2.53%, 1.28%, and 2.67% when using the LGBM, XGB, and RF algorithms, respectively, compared to the results obtained without balancing. Among these, LGBM achieved the highest result with an F1-score of 0.81. Similarly, for model M2, the results show improvements in the high-intensity rain class of 12.72%, 13.46%, and 11.53% for LGBM, XGB, and RF, respectively. Again, LGBM achieved the highest result with an F1-score of 0.62.

For model M3, the F1-score for the moderate rain class improved by 8.47%, 8.47%, and 12.50% when using LGBM, XGB, and RF, respectively. In this case, LGBM and XGB obtained the highest results, both achieving an F1-score of 0.64. For model M4, the F1-score for the very heavy rain class increased by 9.66-fold when using LGBM, resulting in the highest score of 0.29. Using XGB, the F1-score increased by 6.5-fold, while RF achieved a 13-fold improvement, with both algorithms reaching an F1-score of 0.26.

**Table 13**. Performance of rainfall estimation using the different frameworks.

| Rainfall products | CC | MAE | RMSE | mKGE | BIAS |
|---|---|---|---|---|---|
| LGBM-3SC-CF | **0.47** | **2.66** | **5.48** | **0.47** | **1.09** |
| Product based on [5] | 0.44 | 2.95 | 5.77 | 0.41 | 0.83 |

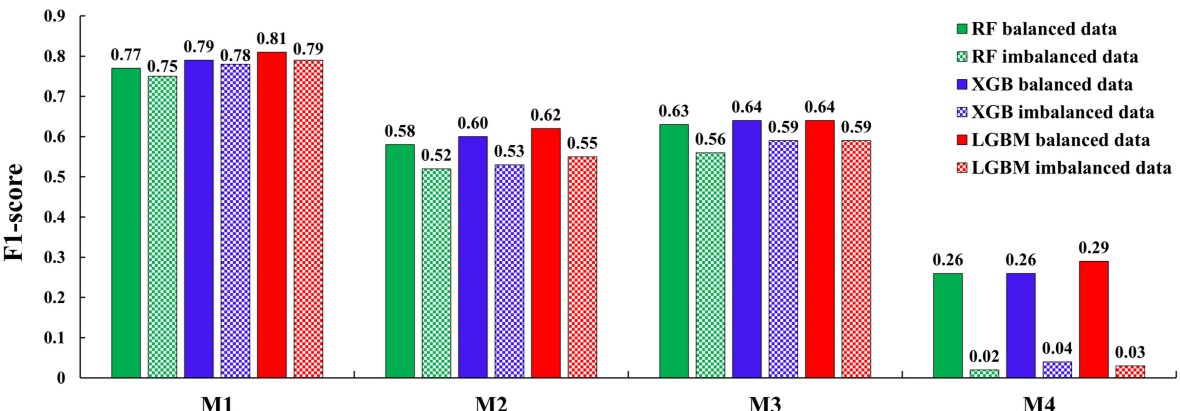

**Fig 14**. Classification results of models M1, M2, M3, and M4, with balanced and imbalanced data.

**Impacts of feature selection techniques.** To evaluate the effectiveness of the feature selection, we compared the performance of the proposed rainfall product, LGBM-3SC-CF with the common feature set, with that of the LGBM-3SC-RF product, which utilizes features selected based on the RF Importance method from the Scikit-learn library [36]. Both rainfall products employ the same architecture with the same LGBM algorithm. Four classification metrics, including POD, CSI, ETS, and HSS, were used for evaluation, and the detailed results are presented in Table 14.

Table 14 shows that the proposed rainfall product, LGBM-3SC-CF, achieved superior results with POD = 0.74, CSI = 0.55, ETS = 0.43, HSS = 0.60, and FAR = 0.30, outperforming the LGBM-3SC-RF product by 1.36%, 3.77%, 7.50%, 5.26%, and 11.76%, respectively, across these metrics. This indicates that the common feature set in this study is well-suited to capturing the actual rainfall patterns in the region, thereby leading to improved rainfall classification performance.

Similarly, the rainfall regression performance of the proposed LGBM-3SC-CF product was also compared with that of the LGBM-3SC-RF product using five metrics (CC, MAE, RMSE, mKGE, and BIAS). Detailed results are presented in Table 15.

Table 15 demonstrates that the proposed rainfall product, LGBM-3SC-CF, achieves higher rainfall regression performance compared to the LGBM-3SC-RF product, which utilizes a feature set selected using the RF importance technique. Specifically, LGBM-3SC-CF yields a CC of 0.47, an MAE of 2.66 mm/h, an RMSE of 5.48 mm/h, an mKGE of 0.47, and a BIAS of 1.09. These values represent respective improvements of 14.63%, 8.28%, 5.35%, 27.03%, and 52.63% over the LGBM-3SC-RF product in terms of CC, MAE, RMSE, mKGE, and BIAS.

Therefore, among all evaluated rainfall products, the proposed LGBM-3SC-CF demonstrates the highest overall performance in both rainfall classification and estimation for the four coastal provinces in the Central region of Vietnam.

**Table 14**. Performance of rainfall estimation using the different input feature sets.

| Rainfall product | POD | CSI | ETS | HSS | FAR |
|---|---|---|---|---|---|
| LGBM-3SC-CF | **0.74** | **0.55** | **0.43** | **0.60** | **0.30** |
| LGBM-3SC-RF | 0.73 | 0.53 | 0.40 | 0.57 | 0.34 |

**Table 15**. Performance of rainfall estimation using the different input feature sets.

| Rainfall products | CC | MAE | RMSE | mKGE | BIAS |
|---|---|---|---|---|---|
| LGBM-3SC-CF | **0.49** | 2.90 | **5.68** | **0.46** | **1.04** |
| LGBM-3SC-RF | 0.41 | 2.90 | 5.79 | 0.37 | 0.81 |

Nevertheless, accurately estimating very high rainfall intensities (>30 mm/h), which commonly occur under complex weather and topographic conditions, remains a significant challenge.

## Conclusion

This study introduces a novel, integrated three-stage classification machine learning architecture designed to significantly enhance the accuracy of rainfall classification and estimation across four coastal provinces in Central Vietnam. Central to this architecture is a specialized cloud masking technique, derived from brightness temperature thresholds of Himawari-8 infrared bands, which substantially improves the performance of both rain/no-rain and low/high-intensity rainfall classification models. Additionally, to optimize multi-class performance and address the severe class imbalance, two data balancing techniques—including sample augmentation for the Range-based Rainfall and Class Weighting—are strategically incorporated into the framework.

The proposed architecture applies three algorithms: RF, XGB, and LGBM, which generate three rainfall products: RF-3SC-CF, XGB-3SC-CF, and LGBM-3SC-CF, respectively. The results demonstrate that among the three proposed rainfall products, LGBM-3SC-CF achieved the highest performance. It significantly outperformed four regional satellite products (IMERG Final Run V07, GSMaP_MVK_Gauge V07, IMERG Early V06, and PERSIANN_CCS) as well as the version without a cloud mask (LGBM-3S) in both classification and regression tasks.

Furthermore, the architecture's design establishes a robust and flexible methodological foundation that allows for future development into near real-time applications through the straightforward substitution of reanalysis data with real-time meteorological features. As the framework is extended to such applications, rare and extreme rainfall events may still influence model performance, particularly in short-term or event-based contexts, highlighting the need for a more detailed assessment of high-intensity events in future work.

## Author contributions

**Conceptualization:** An Hung Nguyen.

**Data curation:** Thanh Thi Nhat Nguyen.

**Formal analysis:** Dong Vu Duy.

**Methodology:** Phat T. Nguyen.

**Software:** Phat T. Nguyen.

**Supervision:** An Hung Nguyen.

**Validation:** An Hung Nguyen, Huyen Thi Nguyen.

**Visualization:** Thanh Thi Nhat Nguyen.

**Writing – original draft:** Dong Vu Duy.

**Writing – review & editing:** Huyen Thi Nguyen.

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
