## [Decision Letter · Decision Letter 0]

23 Sep 2025

PONE-D-25-46623Enhancing rainfall estimation accuracy with machine learning, cloud masking, and multi-source data: a case study of four coastal provinces in central VietnamPLOS ONE

Dear Dr. Nguyen,

Thank you for submitting your manuscript to PLOS ONE. After careful consideration, we feel that it has merit but does not fully meet PLOS ONE’s publication criteria as it currently stands. Therefore, we invite you to submit a revised version of the manuscript that addresses the points raised during the review process.

We look forward to receiving your revised manuscript.

Kind regards,

Mou Leong Tan

Academic Editor

PLOS ONE

Journal Requirements:

3. Please amend the manuscript submission data (via Edit Submission) to include author Hung An Nguyen

4. Please amend your authorship list in your manuscript file to include author Hung Hung Nguyen

5. We note that Figures 1, 5, 7, 8, 11, and 13 in your submission contain map images which may be copyrighted. All PLOS content is published under the Creative Commons Attribution License (CC BY 4.0), which means that the manuscript, images, and Supporting Information files will be freely available online, and any third party is permitted to access, download, copy, distribute, and use these materials in any way, even commercially, with proper attribution. For these reasons, we cannot publish previously copyrighted maps or satellite images created using proprietary data, such as Google software (Google Maps, Street View, and Earth). For more information, see our copyright guidelines: http://journals.plos.org/plosone/s/licenses-and-copyright.

a. You may seek permission from the original copyright holder of Figures 1, 5, 7, 8, 11, and 13 to publish the content specifically under the CC BY 4.0 license.

Additional Editor Comments:

Two reviewers have provided constructive comments to help the authors improve their manuscript. I agree with their suggestions, particularly the need to extend the study period and to conduct a more comprehensive review of the subject matter.

Reviewers' comments:

Reviewer's Responses to Questions

**Comments to the Author**

1. Is the manuscript technically sound, and do the data support the conclusions?

Reviewer #1: Yes

Reviewer #2: Yes

2. Has the statistical analysis been performed appropriately and rigorously?

Reviewer #1: No

Reviewer #2: Yes

3. Have the authors made all data underlying the findings in their manuscript fully available?

Reviewer #1: No

Reviewer #2: No

4. Is the manuscript presented in an intelligible fashion and written in standard English?

Reviewer #1: Yes

Reviewer #2: Yes

5. Review Comments to the Author

Reviewer #1: A. Article Overview

This manuscript describes a machine learning-based methodology designed to significantly improve rainfall estimation accuracy in the four coastal provinces of Central Vietnam, a region characterized by complex topography and climate. The researchers developed an innovative machine learning-based product using a Light Gradient Boosting Machine (LGBM) model fed with multi-source data, including Himawari-8 satellite imagery, ERA5 reanalysis data, ASTER DEM, and ground-based rain gauge measurements.

A key innovation is a three-stage classification architecture that first distinguishes rain from no-rain areas, then classifies rain into low or high intensity, and finally categorizes it into four detailed classes: small, moderate, heavy, and very heavy rain. This process is filtered by two cloud masks generated from brightness temperature thresholds to identify rain-bearing clouds and their intensity, alongside data balancing techniques to address severe class imbalance. The efficacy of the resulting outcome was evaluated against five established regional rainfall products and weather radar imagery.

B. Observations.

1. The model is trained on data from only two years (2019-2020), so from a statistical point of view the period is not relevant. A longer time series is needed, which includes more years with different climatic conditions.

2. The use of cloud masks, while reducing false alarms, also led to a decrease in the probability of detection, and with regard to extreme events the model cannot be considered relevant, an aspect also recognized by the authors.

3. The use of rain gauge data for supervised machine learning training makes the accuracy of the model good but is related to their resolution.

C. General comments

The research seems interesting but it needs to be redone on a statistically relevant series, and I do not necessarily require it to be a WMO reference period, for example 1991-2020, or a statistically valid one of at least 20 years, but it must contain at least 8 years of training, with two years of validation.

Best wishes,

The reviewer.

Reviewer #2: The manuscript presents an approach that integrates a Light Gradient Boosting Machine (LGBM) model with multi-source data to classify and estimate rainfall in four coastal regions of Central Vietnam. Although the topic is relevant and the study contributes to a broader effort to improve the accuracy of rainfall forecasting, the manuscript in its current form requires significant revision to meet standards.

The abstract, for example, contains an excessive amount of detail that obscures the main purpose and contribution of the study. It should be thoroughly revised to be more concise and focused, presenting only the essential elements: the research problem, objectives, a brief description of the proposed approach, the main results, and their significance. Overly detailed methodological explanations and lengthy background information should be removed to ensure that the abstract clearly conveys the essence and relevance of the study to a broad audience.

Similarly, the introduction does not adequately place the study in the context of existing research. It lacks a critical synthesis of previous studies and does not clearly articulate the specific knowledge gaps that the work seeks to address. To strengthen this section, the author should integrate the relevant literature more effectively, highlight the limitations of previous work, and explicitly explain how this study builds on and advances previous efforts. A more thorough and focused literature review would make the novelty and research needs clearer to the reader.

The methodology section also needs significant reorganization. Its current structure is fragmented, and its logical sequence is difficult to follow. Authors are encouraged to begin by outlining the overall methodological framework and data processing steps before explaining the theoretical and algorithmic details of the LightGBM model. Establishing a clearer structure will help the reader understand the study design and the rationale for the chosen approach.

A further limitation is the exclusive reliance on the LightGBM model without comparison with other robust machine learning or statistical models. Without such a benchmark, it is difficult to assess the relative performance or originality of the proposed method. Incorporating additional underlying models such as Random Forest, XGBoost, Support Vector Machines, or appropriate deep learning models would provide a stronger basis for evaluating the effectiveness and specificity of the approach.

In addition, the manuscript does not describe any model optimization or hyperparameter tuning strategies. Since hyperparameter tuning is essential to achieve robust and reproducible performance, authors should clearly state the optimization method used (such as grid search, random search, Bayesian optimization or Optuna) and report the tuned parameters used in the final model. Providing this information is essential to ensure transparency, reproducibility and fairness in model evaluation.

The discussion section would also benefit from a stronger engagement with prior research. It currently lacks critical comparisons with existing studies, which weakens the findings of this study. Incorporating relevant prior findings and discussing how the results align with or deviate from them would help contextualize the study’s contribution and explain how it advances current knowledge in the field.

Finally, the conclusion section should be revised for brevity and focus. It should not repeat detailed findings but instead synthesize the main contributions, outline their implications, and briefly note directions for future work. Although the manuscript addresses an important and timely topic, it requires substantial revision to improve structure and clarity before it can be considered for publication.

6. PLOS authors have the option to publish the peer review history of their article (what does this mean?). If published, this will include your full peer review and any attached files.

Reviewer #1: **Yes:** Cornel Ilinca

Reviewer #2: No

---

## [Author Response · Author response to Decision Letter 1]

8 Dec 2025

We have already responded to every comment from the Editor and Reviewers. The responses are included in the file "Response to Reviewers"

---

## [Decision Letter · Decision Letter 1]

18 Jan 2026

PONE-D-25-46623R1Enhancing rainfall estimation accuracy with machine learning, cloud masking, and multi-source data: a case study of four coastal provinces in central VietnamPLOS One

Dear Dr. Nguyen,

Thank you for submitting your manuscript to PLOS ONE. After careful consideration, we feel that it has merit but does not fully meet PLOS ONE’s publication criteria as it currently stands. Therefore, we invite you to submit a revised version of the manuscript that addresses the points raised during the review process.

We look forward to receiving your revised manuscript.

Kind regards,

Mou Leong Tan

Academic Editor

PLOS One

Journal Requirements:

Additional Editor Comments:

The authors have satisfactorily addressed most of the reviewers' concerns. However, a few minor points still require attention. I recommend a final round of revisions. Thank you.

Reviewers' comments:

Reviewer's Responses to Questions

**Comments to the Author**

1. If the authors have adequately addressed your comments raised in a previous round of review and you feel that this manuscript is now acceptable for publication, you may indicate that here to bypass the “Comments to the Author” section, enter your conflict of interest statement in the “Confidential to Editor” section, and submit your "Accept" recommendation.

Reviewer #1: All comments have been addressed

Reviewer #2: All comments have been addressed

2. Is the manuscript technically sound, and do the data support the conclusions?

Reviewer #1: Yes

Reviewer #2: Yes

3. Has the statistical analysis been performed appropriately and rigorously?

Reviewer #1: Yes

Reviewer #2: Yes

4. Have the authors made all data underlying the findings in their manuscript fully available?

Reviewer #1: Yes

Reviewer #2: Yes

5. Is the manuscript presented in an intelligible fashion and written in standard English?

Reviewer #1: Yes

Reviewer #2: Yes

6. Review Comments to the Author

Reviewer #1: The authors' response to Point 1 is correct. They successfully demonstrated that the model was not overfitted to the initial two-year period, as it maintains consistent performance over the extended five-year dataset.

Regarding Point 2, the potential lack of relevance for extreme events reflects a deliberate methodological choice to optimize global error. However, it should be noted that high-intensity, rare, or localized events could still substantially influence model outcomes.

Regarding Point 3, the map provided confirms that the rain gauge stations offer uniform territorial coverage, ensuring the spatial representativeness and validity of the model's training.

Reviewer #2: The authors have addressed the reviewers’ comments and made the necessary corrections. However, the manuscript should be carefully proofread prior to publication, and the quality of the figures should be further improved to enhance clarity and readability.

7. PLOS authors have the option to publish the peer review history of their article (what does this mean?). If published, this will include your full peer review and any attached files.

Reviewer #1: **Yes:** Cornel Ilinca

Reviewer #2: No

---

## [Author Response · Author response to Decision Letter 2]

26 Jan 2026

Detailed responses to all comments are provided in the uploaded file "Response to Reviewer.docx".

---

## [Editor Report · Decision Letter 2]

29 Jan 2026

Enhancing rainfall estimation accuracy with machine learning, cloud masking, and multi-source data: a case study of four coastal provinces in central Vietnam

PONE-D-25-46623R2

Dear Dr. Nguyen,

We’re pleased to inform you that your manuscript has been judged scientifically suitable for publication and will be formally accepted for publication once it meets all outstanding technical requirements.

Kind regards,

Mou Leong Tan

Academic Editor

PLOS One

Additional Editor Comments (optional):

Most reviewer comments have been addressed, and the manuscript is now suitable for publication in PLOS ONE. The study provides valuable benefits for rainfall estimation in Vietnam.
---

## [Editor Report · Acceptance letter]

PONE-D-25-46623R2

PLOS One

Dear Dr. Nguyen,

I'm pleased to inform you that your manuscript has been deemed suitable for publication in PLOS One. Congratulations! Your manuscript is now being handed over to our production team.

Kind regards,

on behalf of

Dr. Mou Leong Tan

Academic Editor

PLOS One